# SCRQE: Subjective comparative relation quintuple extraction from questions in product domain

**Marzieh Babaali**⊙**, Afsaneh Fatemi**⊙*, **Mohammad Ali Nematbakhsh**

Department of Software Engineering, University of Isfahan, Isfahan, Iran

* A_fatemi@eng.ui.ac.ir

## Abstract

The extraction of subjective comparative relations is essential in the field of question answering systems, playing a crucial role in accurately interpreting and addressing complex questions. To tackle this challenge, we propose the SCQRE model, specifically designed to extract subjective comparative relations from questions by focusing on entities, aspects, constraints, and preferences. Our approach leverages multi-task learning, the Natural Language Inference (NLI) paradigm, and a specialized adapter integrated into RoBERTa_base_go_emotions to enhance performance in Element Extraction (EE), Compared Elements Identification (CEI), and Comparative Preference Classification (CPC). Key innovations include handling X- and XOR-type preferences, capturing implicit comparative nuances, and the robust extraction of constraints often neglected in existing models. We also introduce the Smartphone-SCQRE dataset, along with another domain-specific dataset, Brands-CompSent-19-SCQRE, both structured as subjective comparative questions. Experimental results demonstrate that our model outperforms existing approaches across multiple question-level and sentence-level datasets and surpasses recent language models, such as GPT-3.5-turbo-0613, Llama-2-70b-chat, and Qwen-1.5-7B-Chat, showcasing its effectiveness in question-based comparative relation extraction.

## 1. Introduction

In the field of automatic question answering (AQA), interpreting subjective comparative questions is a complex task. Subjective comparative questions, which are commonly seen on E-commerce platforms, seek to understand links between two or more entities based on individual impressions. For instance, the question: "Is the Samsung A's zoom inferior to the Samsung B in dim lighting?" represents the Quintuple format: Subject Entity, Object Entity, Compared Aspect, Constrained phrase, and Comparative Preference. Not all questions strictly follow this quintuple structure, often having implicit or missing components.

**Data availability statement:** The dataset, along with pertinent scripts and resources, is accessible via our dedicated GitHub repository at https://github.com/mahsamb/SCRQD.

**Funding:** The author(s) received no specific funding for this work.

**Competing interests:** The authors have declared that no competing interests exist.

In this paper, we focus on the detailed analysis of subjective comparative questions as a crucial step in the automatic question answering (AQA) process. While our work is centered on the analysis and extraction of key elements from these questions, this foundational analysis is essential for generating meaningful and actionable answers in subsequent stages of AQA. By accurately identifying and classifying the components of comparative questions, our work provides valuable insights that can be leveraged by other research efforts aimed at answer generation, ultimately assisting users in making informed decisions.

Our model's outputs, particularly the determination of subject and object entities, the identification of compared aspects, the classification of comparative preferences (CPC), and the extraction of constraints, play a pivotal role in shaping the content and form of the answers:

1. **Determining subject/object entities:** By accurately identifying the entities being compared, our model ensures that the answer is directly relevant to the user's query, focusing on the correct entities without ambiguity.

2. **Compared aspects:** The identification of specific aspects under comparison allows the generated answer to be precise and contextually relevant, addressing exactly what the user is interested in, such as performance, design, or price.

3. **Comparative Preference Classification (CPC) of questions:** The CPC process categorizes the nature of the comparison within the question, such as whether one entity is "better," "worse," or "equal" to another, or more nuanced preferences like "significantly better" or "marginally worse." This classification profoundly influences the answer generated in response to a subjective comparative question, as it directly shapes the direction, tone, and content of the answer.

- **Alignment of answer tone with question's intent:** By identifying the specific comparative preference expressed in the question, the CPC helps ensure that the answer's tone aligns with the user's intent. For example, if a question asks whether a product is "significantly better" than another, the answer can affirm this with strong, decisive language if the comparison holds true, or gently refute it with explanations if it does not. This alignment is critical in providing an answer that resonates with the user's expectations and perceptions.

- **Focus on relevant aspects:** The CPC of the question also guides the answer to focus on the aspects most relevant to the user. For instance, if a question compares two smartphones primarily on their camera quality, and the CPC indicates a "worse" comparison, the answer can delve into the specific shortcomings of the camera in one smartphone compared to the other, ensuring that the response is directly relevant to the user's concerns.

- **Handling ambiguities and preferences:** In cases where the CPC reveals ambiguous or mixed preferences in the question (e.g., "Is the camera quality of Phone A better, but its battery life worse than Phone B?"), the answer can

address each aspect separately, providing a balanced view that helps the user understand the trade-offs. This nuanced handling of preferences ensures that the answer is comprehensive and addresses all facets of the user's query.

- **Enhancing decision-making with nuanced comparisons:** The CPC of the question can highlight nuanced preferences, such as a slight advantage in one area versus a significant disadvantage in another. This allows the answer to help the user weigh these factors in their decision-making process. For example, if a question asks whether a laptop is "slightly better in performance but much worse in portability" than another, the answer can guide the user by emphasizing the relative importance of these aspects based on their likely priorities.

- **Customizing answer length and detail:** The CPC can also influence the length and detail of the answer. For instance, a question classified with a "strong preference" might warrant a more detailed explanation to justify this classification, whereas a question with an "equal" classification might receive a more concise response, reflecting the lack of a clear distinction between the compared entities.

4. **Constraint extraction:** Constraints such as price limits or specific conditions (e.g., under low light conditions) are crucial for tailoring the answer to the user's needs, making the information provided more useful and applicable in specific contexts.

Despite the critical role of analyzing subjective comparative questions, there is a notable lack of extensive research in English that adequately addresses their complexities [1–3]. A comprehensive review of the current literature reveals several significant gaps:

1. **Singular comparative focus:** Previous works fail to address questions with multiple comparative relations, assuming that each question consists of only one comparative relation [1–3]. However, real-world scenarios often involve questions with more than one subjective comparative relation. For example: "*Is it true that smartphone A is better in quality than smartphone B and smartphone C, but smartphone B is better in appearance than smartphone A?*" This question contains multiple comparative relations, challenging the previous works' approach.

2. **Ambiguity in preferences:** Existing models lack a robust mechanism to analyze and interpret the numerous ambiguities that subjective questions frequently present, as demonstrated by examples like "*Is it good or bad to upgrade my LG V20 to Samsung Galaxy S7 Edge?*" and "*Which phone has better overall performance, Samsung X or Samsung Y?*".

3. **Intensity and constraint oversight:** The degrees of comparison and contextual constraints vital for questions are inadequately addressed in current methodologies. For example, "*Is it true that the Samsung A has tremendously better capabilities than the Samsung B?*" conveys greater preference intensity than "*Is it true that Samsung A has better capabilities than Samsung B?*" These instances emphasize the need for a more thorough examination of intensity and constraint oversight in existing research.

4. **Current dataset limitations:** As far as we know, there is no publicly available dataset to derive subjective comparative relations from the questions. There is only one small dataset (the dataset from Yu et al. [2]) comprising 35 subjective comparative questions in this area, which is not annotated for either preference intensity or constraints. Also, all the questions in this dataset are single-sentence, and there is at most one comparative relation in each question.

In response to these challenges, our work offers several key contributions:

- **Extraction of multiple comparative relations**: Our model efficiently handles questions containing multiple subjective comparative relations.

- **Novel preference categorization**: We introduce a new classification system for subjective comparative preferences, including previously overlooked XOR-type and X-type preferences.

- **Specialized datasets**: We developed two datasets— Smartphone-SCQRE and Brands-CompSent-19-SCQRE—specifically designed for subjective comparative quintuple extraction, providing a rich resource for training and evaluation. For the preparation of our datasets, we used three annotators and employed Fleiss' Kappa, Weighted Kappa, and Alpha U to measure inter-annotator reliability across entity, aspect, constraint extraction, and subjective comparative relations.

- **Innovative techniques**: We are the first to incorporate constraint and preference intensity extraction within subjective comparative questions, improving contextual relevance in AQA systems.

This paper introduces a multi-phase extraction pipeline leveraging a specialized variant of the RoBERTa architecture, called RoBERTa_base_go_emotions [4]. Unlike the original RoBERTa [5], this model has been fine-tuned on the GoEmotions dataset [6], which comprises 58k Reddit comments annotated for 27 distinct emotional categories plus a neutral category. By leveraging its ability to capture nuanced emotional undertones, our approach gains enhanced sensitivity to subjective expressions of preference, intensities, and user constraints—elements that are especially crucial for subjective comparative questions. Detailed information about the RoBERTa_base_go_emotions model and its application in our study is thoroughly discussed in Section 3.3.1.

By integrating Multi-Task Learning (MTL) for element extraction, comparative element identification, and preference classification, alongside Domain-Specific Adapter-Based Transfer Learning and Natural Language Inference (NLI) techniques, the model efficiently handles complex relations and implicit preferences.

Our model has proven effective across various domains, with evaluations conducted on the Brands-CompSent-19-SCQRE and Smartphone-SCQRE datasets. Experimental results indicate that SCRQE consistently outperformed prompt-based models such as GPT-3.5-turbo-0613, Llama-2-70b-chat, and Qwen-1.5-7B-Chat, particularly excelling in handling non-gradable preferences and complex relational structures. Additionally, our approach outshined sentence-level methods in tests on SemEval 2014 [7], Camera-COQE [8], and CompSent-19 [9], highlighting its robustness and cross-domain applicability.

Following this initial section, Section 2 offers an in-depth literature review, highlighting similarities and differences with our approach. Section 3 introduces our concepts and definitions, dataset, and model for SCRQE. Section 4 outlines our experimental protocols, evaluation metrics, and achieved results. Conclusions and future research avenues are the focus of Section 6.

## 2. Related work

On product review platforms, subjective comparative questions significantly influence user choices. The exploration of Subjective Comparative Relation Extraction (SCRE) has been explored extensively, yet previous research has often leaned more towards sentence-level analyses than questions. This section examines the prominent studies in the field, systematically divided into two main subsections: "Question-level" and "Sentence-level" examinations. To facilitate a comprehensive view, a summary of the related works, categorized by various attributes like language, analysis level, and extraction types, is provided in Section 2.3.

### 2.1. Question-level

Within the SCRE domain, the extraction of comparative information from questions presents a unique set of challenges. Here, we navigate through foundational works that have explored the complex task of interpreting and understanding comparative subtleties embedded within questions.

Pioneering works on question-level SCRE, such as Moghaddam and Ester's AQA system [1], introduced opinion-based question analysis using pattern matching but struggled with multiple comparative relations. Similarly, Yu et al. [2] employed Stanford CRF-NER and SVM for comparative question analysis but faced challenges handling complex

dynamics. Our system improves upon these by using a neural network approach that handles multiple comparative relations more effectively, reducing manual effort in pattern matching and entity recognition. While Yu et al. [2] provided the only relevant question-level dataset, it contains only 35 subjective comparative questions, lacks annotations for preference intensity and constraints, and is limited to single-sentence questions. This restricts its ability to analyze complex comparative interactions. In contrast, our dataset includes a wide range of comparative questions with detailed annotations, offering a richer understanding of comparative dynamics across product categories.

Diverse methodological approaches such as Li et al. [10], which utilize class sequential rules and semantic role labeling, have set a new path for Comparable Entity Extraction. Similarly, REDDY and Mahesh Babu [11] approach, characterized by weakly supervised bootstrapping, makes strides in sequential pattern learning and offers a unique perspective on entity recognition. Compared to these studies, our approach leverages deep learning to automatically infer comparative structures from question data, which improves accuracy and reduces the need for extensive rule-based configurations.

Global research, such as Saelan et al. [12,13] in Indonesia, emphasizes the importance of constraint extraction in comparative questions but faces challenges with some aspects of comparative relations. These studies are among the few that address constraint extraction, but unlike their model, which lacks support for multiple relations in a single question or distinct subject-object identification, our model handles these tasks, improving depth and accuracy. Similarly, Liu et al. [14] analyzed Chinese product competition with a methodology that, while comprehensive, requires intensive manual effort. In contrast, our model automates much of the process, significantly reducing the need for manual input.

### 2.2. Sentence-level

While questions offer their complexities, sentence-level analyses stand as a foundation in the SCRE research field. This subsection examines the significant contributions that have influenced the understanding of comparative relations found within comprehensive sentences, extracting elements, and identifying patterns.

Foundational studies, such as Jindal and Liu [15], introduced a pre-defined set of comparative quintuples for SCRE in English product reviews using CSR and LSR to extract relations from language patterns. Xu et al. [16] advanced this with a two-level CRF model, extracting multiple comparative relations from sources like Amazon and blogs. Arora et al. [17] furthered the field by using a one-layer BiLSTM with GloVe embeddings to extract comparative elements from 27,000 Amazon sentences, demonstrating its superiority over traditional Semantic Role Labeling (SRL). Our model builds on these advances by utilizing the transformer-based RoBERTa_base_go_emotions architecture, fine-tuned for complex natural language tasks, to discern and classify nuanced comparative preferences with high precision across varied linguistic contexts.

Recent developments include the *UniCOQE* (Unified Comparative Opinion Quintuple Extraction) framework proposed by Yang et al. [18]. This model addresses the limitations of pipeline-based methods by employing a unified generative approach for comparative quintuple extraction. It innovatively treats multiple quintuples as an unordered set, using a set-matching strategy to overcome order bias during training. While *UniCOQE* focuses on the COQE task and shows improvement over prior state-of-the-art methods, our SCRQE model builds on this by incorporating constraint extraction and intensity preference analysis, expanding the scope of comparative relation extraction to better handle contextual and preference-based subtleties.

Liu et al. [8] further advanced the field with the Comparative Opinion Quintuple Extraction (COQE) task, using a multi-stage neural network to identify multiple relations within a single sentence. For instance, as shown in Table 1, in the sentence, "G6 has a worse zoom than G7, but G6's battery is more reliable than G7," the model extracts two quintuples, delineating <Subject Entity, Object Entity, Comparative Aspect, Comparative Opinion, and Comparative Preference>. Our model extends this by integrating Constraint Extraction and intensity preference analysis, offering more nuanced insights into preferences and contexts.

**Table 1. COQE task example [8].**

| | Sentence | G6 has a worse zoom than G7, but G6's battery was more reliable than G7. |
|---|---|---|
| Example 1 | Subject Entity: | G6 |
| | Object Entity: | G7 |
| | Comparative Aspect: | {zoom, battery} |
| | Comparative Opinion: | {worse, more reliable} |
| | COQE: | {(G6, G7, zoom, worse, Worse), (G6, G7, battery, more reliable, Better)} |

Liu et al. [8] also introduced three datasets—Car-COQE, Ele-COQE, and Camera-COQE—with the first two in Chinese and the last one in English, catering to specific product categories within those languages.

Gao et al. [19] introduced an efficient end-to-end COQE solution, enhancing encoding, decoding, and learning by incorporating syntactic dependency features and dynamic structural pruning (DSP) to refine syntax structures. Their non-autoregressive decoding scheme generates quintuples in parallel, improving speed and performance by 2–3 points over baseline models. However, DSP's complexity, dependency feature reliance, and high computational demands may limit its use in resource-constrained settings. In contrast, our approach includes Constraint Extraction and intensity preference analysis, addressing aspects not covered in Gao et al.'s work.

International research has advanced comparative analysis significantly. Wang et al. [20] proposed a hybrid model for Chinese restaurant reviews, assigning weights to comparative relations based on preference intensity, offering nuanced customer insights. Liu, Wang, and Shao [21] later adopted a BERT-CRF model for quintuple extraction, enhancing Comparative Sentences Recognition (CSR) and Comparative Element Extraction (CEE) through advanced language models and sequence labeling. Our SCRQE model, built on a deep learning pipeline, goes further by integrating constraint recognition and intensity preference analysis, elements absent in these models. Leveraging a fine-tuned RoBERTa_base_go_emotions framework, our approach provides more granular comparative analysis and deeper understanding of context and preferences.

CPC research has advanced through various methods. Younis et al. [22] applied machine learning classifiers for multi-class CPC across four datasets, with Random Forest performing best (see Table 2). Unlike their approach, our model integrates NLP techniques with a fine-tuned RoBERTa_base_go_emotions model to classify a wide range of comparative preferences directly from text. This allows dynamic adaptation to different contexts without manual entity identification or dataset dependence. Our approach reduces manual labor while enhancing accuracy and scalability, automating the extraction and classification of comparative elements for greater precision in detecting nuanced comparative relations across diverse contexts.

Panchenko et al. [9] tackled the CPC task with the CompSent-19 dataset using an XGBoost model and InferSent embedding (see Table 3). In contrast, our SCRQE model leverages the RoBERTa_base_go_emotions transformer with a multi-layered approach, improving both depth and accuracy in preference classification. By integrating RoBERTa_base_go_emotions with a sentence-pair classification framework, our model outperforms traditional techniques, enabling more precise analysis of comparative sentences. This enhances the recognition of preferences across complex structures and varied expressions. Additionally, our deep learning approach expands the ability to classify nuanced preferences, setting a new standard for precision and adaptability in CPC.

Recent advancements include Kang et al.'s work on LLM-augmented Preference Learning from Natural Language [23], which explores the application of large language models (LLMs) for Comparative Preference Classification (CPC). This study investigates the effectiveness of models like GPT-3.5 and LLaMa-2 in handling complex and nuanced preferences from longer texts, such as those in the College Confidential and CompSent-19 datasets. Their findings show that LLMs outperform state-of-the-art methods for long and multi-sentence texts, showcasing the potential of few-shot learning

**Table 2. Sentences and associated preference labels [22].**

| # | Sentence | Preference Label Assignment |
|---|----------|----------------------------|
| 1 | *I like android but also likes iOS.* | *neu_neu* |
| 2 | *Android is better than iOS.* | *pos_neg* |
| 3 | *I own android and iOS both are fantastic operating systems.* | *pos_pos* |
| 4 | *I have used both android and iOS but i prefer android.* | *neu_pos* |

**Table 3. Sample sentences from CompSent-19 dataset [9], with preference indications. Note: Sequence matters—preferences reference the initial entity in comparison to the subsequent one.**

| Domain | Sentence | Preference Label Assignment |
|--------|----------|----------------------------|
| Brands | *Honda quality has gone downhill, Hyundai or Ford is a much better value.* | *Worse* |
| CompSci | *I've concluded that it is better to use Python for scripting rather than Bash.* | *Better* |
| Random | *I've grown older and wiser and avoid the pasta and bread like the plague.* | *None* |

to improve preference detection without extensive fine-tuning. In line with this, our work leverages transformer-based models like RoBERTa_base_go_emotions but extends it with task-specific optimizations for subjective comparative relations, including Constraint Extraction and Preference Intensity Classification, which were not directly addressed in Kang et al.'s framework. This allows for deeper comparative analysis across different product categories, particularly in Smartphone-SCQRE and Brands-CompSent-19-SCQRE datasets, offering enhanced precision in handling complex queries.

Our dataset improves significantly on CompSent-19 by covering a broader range of comparative questions. Unlike CompSent-19, which focuses solely on Comparative Preference Classification (CPC) with two entities per sentence, our dataset supports multiple preferences per sentence and includes detailed annotations on preference intensity and constraints. This allows for richer, more accurate analysis of comparative relations, including aspects alongside entities. By incorporating diverse linguistic patterns and supporting multilingual evaluations, our dataset greatly extends the scope of comparative analysis in natural language processing.

Ma et al. [24] advanced the field with the Entity-aware Dependency-based Deep Graph Attention Network, using the CompSent-19 dataset [9] to achieve state-of-the-art results. While their approach required manual entity identification, our model automates the entire process. By integrating RoBERTa_base_go_emotions's transformer architecture, our model enables real-time entity and preference detection, reducing manual effort and improving scalability and precision. This provides a more adaptable and efficient solution for handling complex comparative data across various domains.

## 2.3. Summary of the related works

The various efforts presented in this section are summarized in Table 4. They are categorized based on the language of the study, levels of analysis (sentence-level or question-level), types of Comparative Element Extraction (CEE)—which includes entity, subject/object identification, aspect, and constraint—, Comparative Preference Classification (CPC) types, whether basic or intensity-focused, and types of SCRE, specifically indicating if they address multiple relations.

## 3. Model overview

This section presents the proposed pipeline, specifically designed for extracting comparative relations from subjective comparative questions. We begin the discussion with foundational concepts and definitions related to question-level comparative relation extraction. Subsequent segments offer a brief overview of the Smartphone-SCQRE and Brands-CompSent-19-SCQRE datasets, emphasizing their role as key resources in training and validating our model, showcasing

**Table 4. Overview of the related works.**

| Author(s) | Language | Level | | CEE | | | | CPC | | SCRE |
|---|---|---|---|---|---|---|---|---|---|---|
| | | Question | Sentence | Entity | Subject/ object | Aspect | Constraint | Basic | Intensity | Multi- ple (>=1) |
| Jindal and Liu [15] | English | | ✓ | ✓ | ✓ | ✓ | | ✓ | | |
| Xu et al [16]. | English | | ✓ | ✓ | ✓ | ✓ | | ✓ | | ✓ |
| Moghaddam, and Ester [1] | English | ✓ | | ✓ | | ✓ | | ✓ | | |
| Yu et al [2]. | English | ✓ | | ✓ | | ✓ | | ✓ | | |
| Li et al [10]. | English | ✓ | | ✓ | | | | | | |
| Saelan et al [12]. | Indonesian | ✓ | | ✓ | | ✓ | ✓ | | | |
| Wang et al [20]. | Chinese | | ✓ | ✓ | ✓ | ✓ | | ✓ | ✓ | ✓ |
| Arora et al [17]. | English | | ✓ | ✓ | ✓ | ✓ | | | | |
| Saelan et al [13]. | Indonesian | ✓ | | ✓ | | ✓ | ✓ | ✓ | | |
| Younis et al [22]. | English | | ✓ | | | | | ✓ | | |
| Panchenko et al [9]. | English | | ✓ | | | | | ✓ | | |
| Ma et al [24]. | English | | ✓ | | | | | ✓ | | |
| Liu et al [21]. | Chinese | | ✓ | ✓ | ✓ | ✓ | | | | |
| Liu et al [14]. | Chinese | ✓ | ✓ | ✓ | ✓ | ✓ | | ✓ | ✓ | ✓ |
| Liu et al [8]. | English, Chinese | | ✓ | ✓ | ✓ | ✓ | | ✓ | | ✓ |
| REDDY and Mahesh Babu [11] | English | ✓ | | ✓ | | | | | | |
| Gao et al [19]. | English, Chinese | | ✓ | ✓ | ✓ | ✓ | | ✓ | | ✓ |

the diversity and complexity of subjective comparative questions they contain. Next, we explore further the model architecture and its complexities.

### 3.1. Concepts and definitions

To set the foundation for the upcoming discussion, we present key definitions relevant to comparative relations in subjective comparative questions.

**Definition 1-Entity:** An entity is a product or its brand, like "*Samsung Galaxy A51*" or "*Samsung*".

**Definition 2-Aspect:** As defined by Kang [25], an aspect is a specific characteristic or feature of an entity. It can be explicit (directly mentioned) or implicit (indirectly referred to), as described by Cruz et al. [26]. For instance, "*gaming experience*" in "*Which smartphone offers a better gaming experience, Samsung A or Samsung B?*" is an explicit aspect, while "heavy" implies the implicit aspect "weight".

**Definition 3-Compared entity:** This is an entity that is compared to another in a question, either as a Subject or Object Entity. Notably, entities in a comparative question aren't always comparative. For instance, "*Samsung A*" in "*I would like to buy a Samsung A. Is Samsung B better than Samsung C?*" isn't a compared entity.

**Definition 4-Compared aspect:** The explicit attribute or feature over which entities are being compared. Similarly, not all aspects in a question are necessarily comparative.

**Definition 5-Constraint:** Constraints or conditions limit the scope of potential answers. For example, in "*Which Samsung smartphone is better than iPhone 12 under 600 $?*", the constraints are "*Samsung*" and "*under 600 $*".

**Definition 6-Subjective comparative questions:** These are questions about personal perspectives on the comparison of entities concerning a particular aspect, e.g., "*Why is Samsung A better than Samsung B?*".

**Definition 7-Objective comparative questions:** These are reality-based questions about the comparison of entities, e.g., "*Is Samsung A's screen bigger than Samsung B's?*".

**Definition 8-Comparative relation:** Expressed as a 5-Tuple: (Subject Entity, Object Entity, Compared Aspect, Constraint, Comparative Preference), it describes the relation between two sets of compared entities, aspects, and constraints. Not every element is always present.

**Definition 9-Comparative preference:** It indicates the direction of comparison and can be Gradable (ranking entities) or Non-Gradable (highlighting similarities or differences without ranking). Gradable comparisons can be Non-Equative (like "better" or "worse") or Equative (like "same" or "similar"). For simplification, we group Simple-Equative and Quasi-Equative under Equative (=). Non-Gradable examples include terms like "distinguishable" or "different". The basic directions are Better (>), Worse (<), Equal (=), and Non-Gradable (~). For an overview of comparative preference types and examples, see Table 5.

### 3.2. Introduction to the datasets

This section discusses the Smartphone-SCQRE and Brands-CompSent-19-SCQRE datasets, designed to enhance research in extracting subjective comparative relations for question analysis systems. The following subsections provide an overview of these datasets, detailing the selection and annotation process, and explain the rigorous methods employed to ensure the quality and integrity of the data across all three datasets, as well as the details of other sentence-level datasets used for comparison.

#### 3.2.1. Dataset overview.
To the best of our knowledge, the only available dataset related to SCRE from comparative questions was provided by Yu et al. [2], containing 220 mobile domain questions. Of these, only 35 are subjective comparative and lack constraint and preference intensity annotations. To address this shortage, we introduced two datasets: the "Smartphone-SCQRE" dataset and the "Brands-CompSent-19-SCQRE" dataset.

The Smartphone-SCQRE dataset consists of 2,300 manually annotated subjective comparative questions collected from sources like Quora. The "Brands-CompSent-19-SCQRE" dataset contains 927 questions, adapted from the CompSent-19 dataset, and was selected to evaluate the robustness and generalizability of our model in handling subjective comparative questions from a different domain. We processed the subjective comparative sentences in this dataset, converted them into question format, and applied the same quintuple annotation structure as Smartphone-SCQRE, including the subject entity, object entity, compared aspect, constraint, and preference. Since the number of sentences from that domain meeting our criteria for subjective comparative forms was low, in some cases, we generated multiple questions from one sentence, using various subjective comparative forms—including target, yes/no, attitude, and polarity—as outlined in Babaali et al. [27], with different comparative preferences.

For both datasets, three annotators, experienced in sentiment analysis and proficient in English, labeled each question, with the majority vote determining the final tags. The dataset statistics for CEE, CPC, and SCRE tasks are provided in Tables 6–8 for both datasets.

Since our work is the first to extract subjective comparative quintuples from questions, there was no directly comparable dataset available for a comprehensive evaluation; therefore, we also used sentence-level related datasets for evaluation. Given the absence of relevant question-level datasets, we incorporated three sentence-level benchmarks: Camera-COQE [8], SemEval 2014 [7], and CompSent-19 [9]. Their characteristics are detailed in Tables 9–11. Camera-COQE is a subset of COQE, with 20% of its sentences containing multiple opinion comparative relations.

The SemEval 2014 dataset was employed for the Aspect Extraction task, focusing on product and service reviews, with annotated aspect terms provided for each sentence.

Additionally, the CompSent-19 dataset was used for evaluating the CPC task. As the first dataset designed for CPC, CompSent-19 spans three domains and follows an 80:20 train-test split. Each sentence includes entity-pair annotations and comparative preference labels, though only 27% of the sentences contain comparative relations.

**Table 5. Comparative preference types with examples.**

| Comparative Preference Types | | | Sample Question |
|---|---|---|---|
| Gradable | Non-Equative | Better | Is the iPhone X's appearance better than the Samsung Y? |
| | | | Which is the best smartphone under 1000 $? |
| | | Worse | Is the iPhone X's appearance worse than the Samsung Y? |
| | | | Which is the worst smartphone under 1000 $? |
| | Equative | Simple Equative | Which phone has the exact (same) quality as the Samsung Y? |
| | | | Is the performance of the iPhone X identical to Samsung Y? |
| | | Quasi Equative | Which phone is most similar to iPhone X? |
| | | | Is iPhone X's appearance is as good as Samsung Y? |
| | | | Which smartphone appearance is similar to the OnePlus 7T phone? |
| | | | Is the Samsung A almost the same quality as the latest iPhone B? |
| Non-Gradable | | | How does the user experience vary between the OnePlus 8T and the Google Pixel 5? |
| | | | Is the Realme 8i in competition with Samsung A50? |
| | | | Is the quality of the iPhone X different from Samsung Y? |
| | | | Which iPhone series appearance looks distinguishable from the others? |
| | | | What is the replacement phone for my Xiaomi Redmi Note 10 Lite among Oppo A54 and Vivo Y20? It must cost less than $1000. |

**Table 6. Statistics for the Smartphone-SCQRE and Brands-CompSent-19-SCQRE Datasets Regarding the CEE Task.**

| Elements | Info | Smartphone-SCQRE | Brands-CompSent-19-SCQRE |
|---|---|---|---|
| Questions | # Questions | 2300 | 927 |
| Entity | # Entities | 5802 | 2900 |
| | # Single-Word Entities | 595 | 1910 |
| | # Multi-Word Entities | 4817 | 990 |
| | Max Multi-Word | 9 | 7 |
| | Subject Entity | 3629 | 1565 |
| | Object Entity | 1783 | 1335 |
| Aspect | # Aspects | 2557 | 1052 |
| | # None-Generalized "features" | 2006 | 634 |
| | # Generalized "features" | 551 | 418 |
| | # Single-Word Aspects | 577 | 402 |
| | # Multi-Word Aspects | 1429 | 232 |
| | Max Multi-Word | 7 | 5 |
| Constraint | # Constraints | 301 | 268 |
| | # Single-Word Constraints | 39 | 25 |
| | # Multi-Word Constraints | 262 | 243 |
| | Max Multi-Word | 13 | 12 |

**Table 7. Statistics for the Smartphone-SCQRE and Brands-CompSent-19-SCQRE Datasets Concerning the CPC Task.**

| Preference Type | Smartphone-SCQRE | Brands-CompSent-19-SCQRE |
|---|---|---|
| Better | 331 | 296 |
| Strong Better | 275 | 116 |
| Equal | 162 | 53 |
| Worse | 156 | 72 |
| Strong Worse | 112 | 53 |
| XOR-Strong Better | 129 | 53 |
| XOR-Better | 271 | 41 |
| XOR-Equal | 122 | 45 |
| XOR-Worse | 134 | 53 |
| XOR-Strong Worse | 220 | 47 |
| X-Strong Better | 128 | 55 |
| X | 202 | 60 |
| X-Strong Worse | 129 | 52 |
| Non-Gradable | 189 | 56 |
| **Total** | **2560** | **1052** |

**Table 8. Statistics for the Smartphone-SCQRE and Brands-CompSent-19-SCQRE Datasets Related to the SCRE Task.**

| Info | Smartphone-SCQRE | Brands-CompSent-19-SCQRE |
|---|---|---|
| # One-Relation Questions | 2058 | 821 |
| # Two-Relation Questions | 229 | 93 |
| # Three-Relation Questions | 12 | 7 |
| # Four-Relation Questions | 1 | 6 |

**Table 9. Data insights from the Camera-COQE dataset [8].**

| Info | | # |
|---|---|---|
| **#Sentence** | #Comparative | 1705 |
| | #Non-Comparative | 1599 |
| | #Multi-Comparisons | 500 |
| | #Comparison Per Sentence Percentage | 1.4 29.3% |
| **#Element** | Subject Entity | 1649 |
| | Object Entity | 1316 |
| | Comparative Aspect | 1368 |
| | Comparative Preference | 2442 |

In Table 6, **#** Generalized 'features' refers to explicit aspects mentioned in the question, while **#** None-Generalized 'features' applies to cases where no specific aspect is present, and a generalized "features" label is assigned.

**3.2.2. Data quality and integrity.** Maintaining high standards for data quality and integrity is crucial for the creation and effectiveness of the dataset. This section discusses the methods used to develop a dataset that supports the needs

**Table 10. Dataset statistics for SemEval 2014 (based on Augustyniak et al. [28]).**

| | Info | Laptop | Restaurant |
|---|---|---|---|
| **Train** | # Sentences | 3045 | 3041 |
| | # Aspects | 2358 | 3693 |
| | % Multi-Word Aspects | 37 | 25 |
| **Test** | # Sentences | 800 | 800 |
| | # Aspects | 654 | 1134 |
| | % Multi-Word Aspects | 44 | 28 |
| **All** | # Sentences | 3841 | 3841 |
| | # Aspects | 4827 | 4827 |

**Table 11. Features of the CompSent-19 dataset [9].**

| | | Preference | | | Total |
|---|---|---|---|---|---|
| | | **Better** | **Worse** | **None** | |
| **Domain** | CompSci | 581 | 248 | 1,596 | 2,425 |
| | Brands | 404 | 167 | 1,764 | 2,335 |
| | Random | 379 | 178 | 1,882 | 2,439 |
| | **Total** | **1,364** | **593** | **5,242** | **7,199** |

of NLP research effectively. By employing thorough dataset preparation techniques, incorporating distinctive attributes, implementing a meticulously planned annotation procedure, and applying rigorous criteria for assessing inter-annotator consistency, we adopt a strategy that emphasizes accuracy, dependability, and ethical practices in constructing the dataset.

**Dataset creation: detailed preparation methods and specifications;** This section offers a thorough explanation of the careful steps and criteria involved in creating the Smartphone-SCQRE and Brands-CompSent-19-SCQRE datasets, which is fundamental to its construction. It delves into the methodology used, the selection and modification process of data, and the strategic planning behind compiling the dataset to guarantee its integrity and usefulness for intended research purposes.

We began by assembling data from various sources. For Smartphone-SCQRE, we focused on user contributions from platforms like Amazon (www.amazon.com) and Quora (www.quora.com), with a specific emphasis on the smartphone domain. We also prepared questions by altering existing ones and converting relevant sentences into a question format that met our research criteria. The revised questions were validated by experts in the smartphone field to ensure alignment with our research goals. We employed a multi-faceted approach to minimize bias in the Smartphone-SCQRE dataset by gathering questions from diverse online sources, ensuring a wide range of question styles and domains, which reduced selection bias. We balanced questions across various categories within each domain to mitigate content bias, and detailed annotation guidelines along with calibration sessions helped reduce annotator bias. All data collection and modification processes strictly adhered to platform terms, with ethical considerations, such as protecting personal information and ensuring privacy, as top priorities.

Additionally, Brands-CompSent-19-SCQRE focuses on brand comparisons extracted from the CompSent-19 dataset. Each dataset was carefully curated to ensure that the questions were subjective comparative in nature, clearly specified the entities being compared, and fit within our research categories.

**Annotation process and inter-annotator agreement analysis;** This study utilized the expertise of annotators with backgrounds in computational linguistics and domain-specific knowledge (smartphones and brands). To quantify the

agreement among annotators, we employed various inter-annotator agreement (IAA) metrics. These metrics were applied across all datasets (Smartphone-SCQRE and Brands-CompSent-19-SCQRE):

- **Fleiss' Kappa** [29]: This metric was chosen to assess the agreement on the identification of entities and aspects, which are foundational to constructing the comparative relations. Given its suitability for categorical data and its accommodation of multiple raters, Fleiss' Kappa provided a robust measure of consensus for these critical components across the datasets, including Smartphone-SCQRE and Brands-CompSent-19-SCQRE.

- **Weighted Kappa** [30]: To account for the ordinal nature of comparative preferences, encompassing categories like "Better" to "XOR-Strong Worse," Weighted Kappa was employed. This metric assigns different weights to disagreements based on the severity within preference categories, offering a more nuanced assessment of annotator consistency in this intricate judgment domain.

- **Alpha U** [31]: To evaluate the agreement on extracting entire comparative relations, including Subject Entity, Object Entity, Compared Aspect, Constraint, and Comparative Preference, Alpha U was chosen. This metric is an adaptation of Krippendorff's Alpha [32] specifically designed for tasks involving segmenting and classifying text. By employing Alpha U, the study accurately measured annotator agreement on both the identification and contextual classification of these relations.

Table 12 shows the inter-annotator agreement metrics for the datasets Smartphone-SCQRE and Brands-CompSent-19-SCQRE. These metrics, including Fleiss' Kappa, Weighted Kappa, and Alpha U, provide insights into the consistency and reliability of annotations across different comparative relation extraction tasks, such as Entity Identification, Aspect Extraction, Constraint Extraction, Comparative Entity Identification (CEI), Comparative Preference Classification (CPC), and Subjective Comparative Relation Extraction (SCRE).

The analysis of Smartphone-SCQRE and Brands-CompSent-19-SCQRE based on task performance reveals some key insights. In terms of Entity and Aspect Extraction, Smartphone-SCQRE leads slightly with Fleiss' Kappa values of 0.93 and 0.90, respectively. This suggests that smartphones, with their well-defined and distinct features, are easier for annotators to consistently identify compared to brands. The Brands-CompSent-19-SCQRE dataset follows closely behind, with slightly lower scores, indicating a moderate level of agreement among annotators. However, the Constraint Extraction task sees a dip across both datasets, with Smartphone-SCQRE scoring 0.75, which is still higher than the score for Brands-CompSent-19-SCQRE. This reflects the challenge of identifying specific constraints in subjective comparisons, especially when the constraints vary based on user scenarios.

For Comparative Entity Identification (CEI), Smartphone-SCQRE maintains the highest Alpha U score of 0.85, demonstrating its superior consistency in identifying subject-object comparisons. The Comparative Preference Classification (CPC) scores, measured using Weighted Kappa, also highlight Smartphone-SCQRE's stronger performance (0.78), followed by Brands-CompSent-19-SCQRE (0.76), likely due to the broader and more abstract nature of brand comparisons,

**Table 12. Inter-Annotator Agreement Metrics across the Smartphone-SCQRE and Brands-CompSent-19-SCQRE Datasets.**

| Metric | Dataset | Entity | Aspect | Constraint | CEI | CPC | SCQRE |
|---|---|---|---|---|---|---|---|
| **Fleiss' Kappa** | Smartphone-SCQRE | 0.93 | 0.90 | 0.75 | – | – | – |
| | Brands-CompSent-19-SCQRE | 0.92 | 0.89 | 0.73 | – | – | – |
| **Weighted Kappa** | Smartphone-SCQRE | – | – | – | – | 0.78 | – |
| | Brands-CompSent-19-SCQRE | – | – | – | – | 0.76 | – |
| **Alpha U** | Smartphone-SCQRE | – | – | – | 0.85 | – | 0.82 |
| | Brands-CompSent-19-SCQRE | – | – | – | 0.84 | – | 0.80 |

making preference classification more challenging. Finally, in Subjective Comparative Relation Extraction (SCRE), Smartphone-SCQRE again leads with an Alpha U score of 0.82, though the gap between it and Brands-CompSent-19-SCQRE is less pronounced, with the latter scoring 0.80. This suggests that while Smartphone-SCQRE is more consistent overall, the complexity of comparative relations in the brand dataset presents unique challenges in integrating all elements into a coherent relation.

### 3.3. Proposed model

In this section, the Subjective Comparative Quintuple Relation Extraction (SCQRE) model is introduced, which follows a structured pipeline to process subjective comparative questions. This pipeline is visually depicted in Fig 1, outlining the high-level stages involved in extracting and classifying subjective comparative relations.

The SCRQE is organized in three primary stages:

• **Element Extraction (EE):** Extracts entities, aspects, and constraints from the given subjective comparative question.

• **Compared Elements Identification (CEI):** Identifies the roles of the entities and determines a 4-tuple comparative relation for each question: <Subject Entity, Object Entity, Compared Aspect, Constraint>. This stage involves:

◦ **Entity Role Identification (ERI):** Assigns subject and object roles to entities within the question.

◦ **Aggregation:** Combines and filters the entities to form the 4-tuple.

• **Comparative Preference Classification (CPC):** Determines the preference between entities in relation to their mutual aspect, producing a 5-tuple: <Subject Entity, Object Entity, Compared Aspect, Constraint, Comparative Preference>.

For a detailed breakdown of these stages, refer to Fig 2: Block Diagram for SCQRE, which provides a step-by-step representation of the entire process from Element Extraction (EE) to Comparative Preference Classification (CPC). This diagram illustrates how each stage interacts and progresses towards the final quintuple output.

Table 13 further illustrates the outcomes of the EE, CEI, and CPC stages with specific examples. This table demonstrates how the SCQRE model processes real-world questions and extracts the relevant entities, aspects, and comparative preferences.

The proposed approach utilizes the power of the RoBERTa_base_go_emotions model, performing EE, CEI, and CPC tasks concurrently to extract comparative relations. The EE task is designed as three sequence-tagging sub-tasks: Entity, Aspect, and Constraint tagging, using the BIO-tagging scheme to convert the question into a sequence, marking each token as "Beginning", "Inside", or "Outside". The CIE and CPC tasks are structured as sentence pair classification challenges. Subsequent sections will further unravel the details of the pipeline.

#### 3.3.1. Justification for Using RoBERTa_base_go_emotions.
The roberta-base-go_emotions (available at https://huggingface.co/SamLowe/roberta-base-go_emotions) [4] model was chosen for this task due to its robust ability to handle emotional nuances in subjective comparative questions. Unlike the original RoBERTa [5], which was not specifically fine-tuned for emotion-related tasks, roberta-base-go_emotions has been fine-tuned on the GoEmotions dataset [6], a collection of over 58,000 English-language Reddit comments annotated for 28 distinct emotional categories (27 emotions plus a neutral category). This crucial distinction enables the model to detect and assign multiple emotional labels to a single text input, making it especially effective for tasks where subjective and emotionally nuanced comparisons are central. Developed by SamLowe, this specialized variant inherits the Transformer architecture of the roberta-base model while integrating multi-label classification capabilities to capture subtle emotional undertones.

The roberta-base-go_emotions model is built upon the RoBERTa-base architecture, which features a 12-layer Transformer, 12 self-attention heads, a 768-dimensional hidden layer, and around 125 million parameters. Unlike BERT, RoBERTa introduced dynamic masking, removed the next sentence prediction (NSP) task, and leveraged a

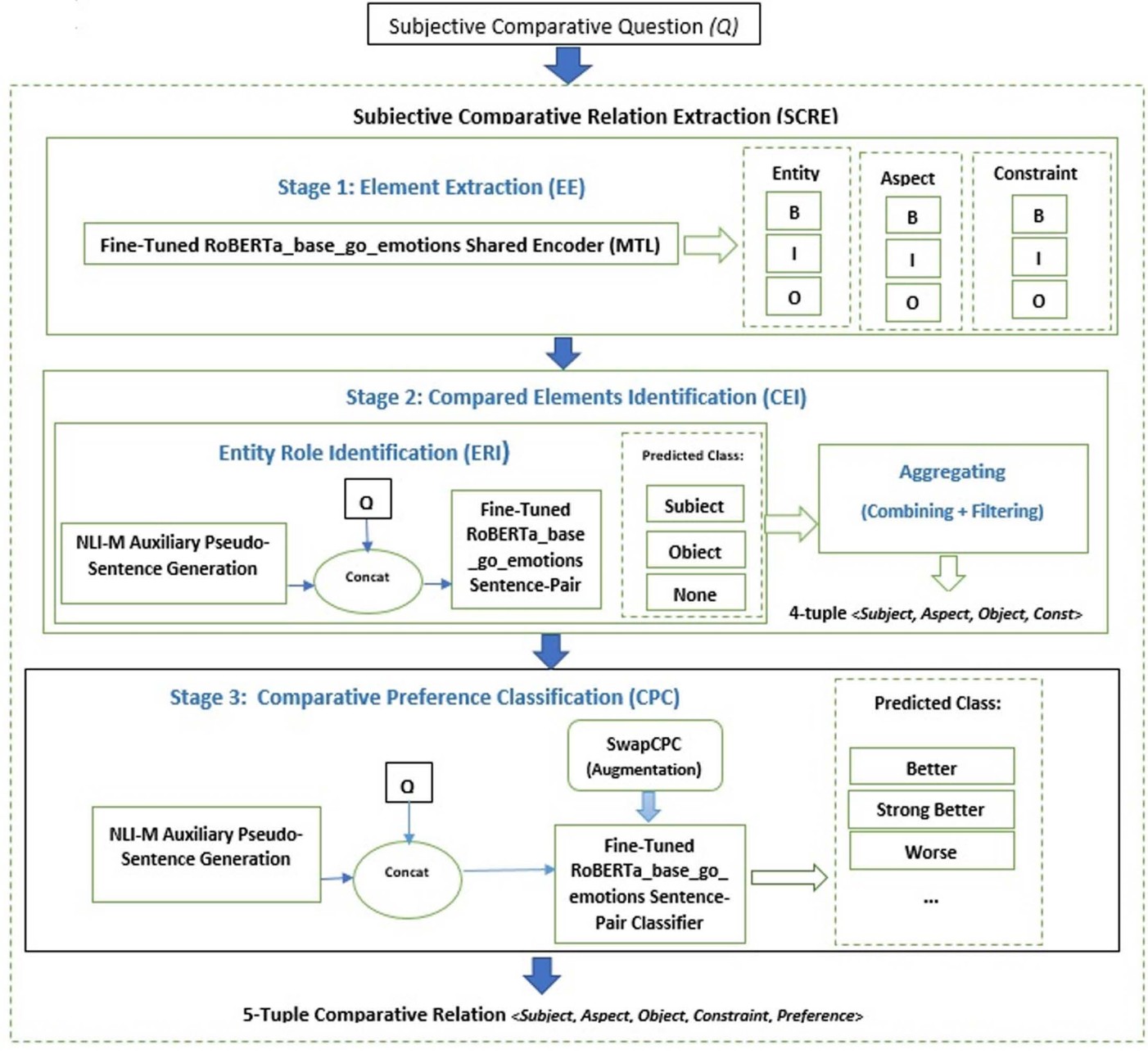

**Fig 1. Pipeline for the Proposed SCRQE model.**

pretraining corpus approximately ten times larger than that of BERT (i.e., 160 GB vs. 16 GB). As a result of this extensive pretraining, RoBERTa-base yields robust representations well suited for a variety of complex NLP tasks. The roberta-base-go_emotions variant then fine-tunes this architecture on the GoEmotions dataset—over 58,000 Reddit comments annotated for 28 emotional categories—enabling it to capture nuanced emotional undertones critical for tasks involving subjective and emotionally charged content.

**Subjective Comparative Question:**

I want to buy the iPhone 14 because of its excellent zoom features, but it has a significantly large size. Excluding that, does the Google Pixel 6a offer better camera quality than the Asus Zenfone 8 while also supporting 5G?

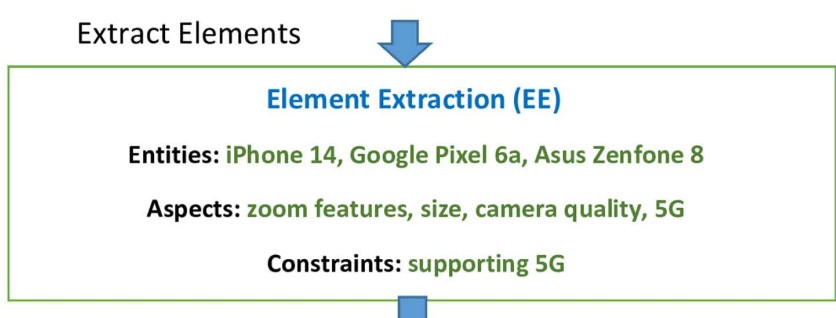

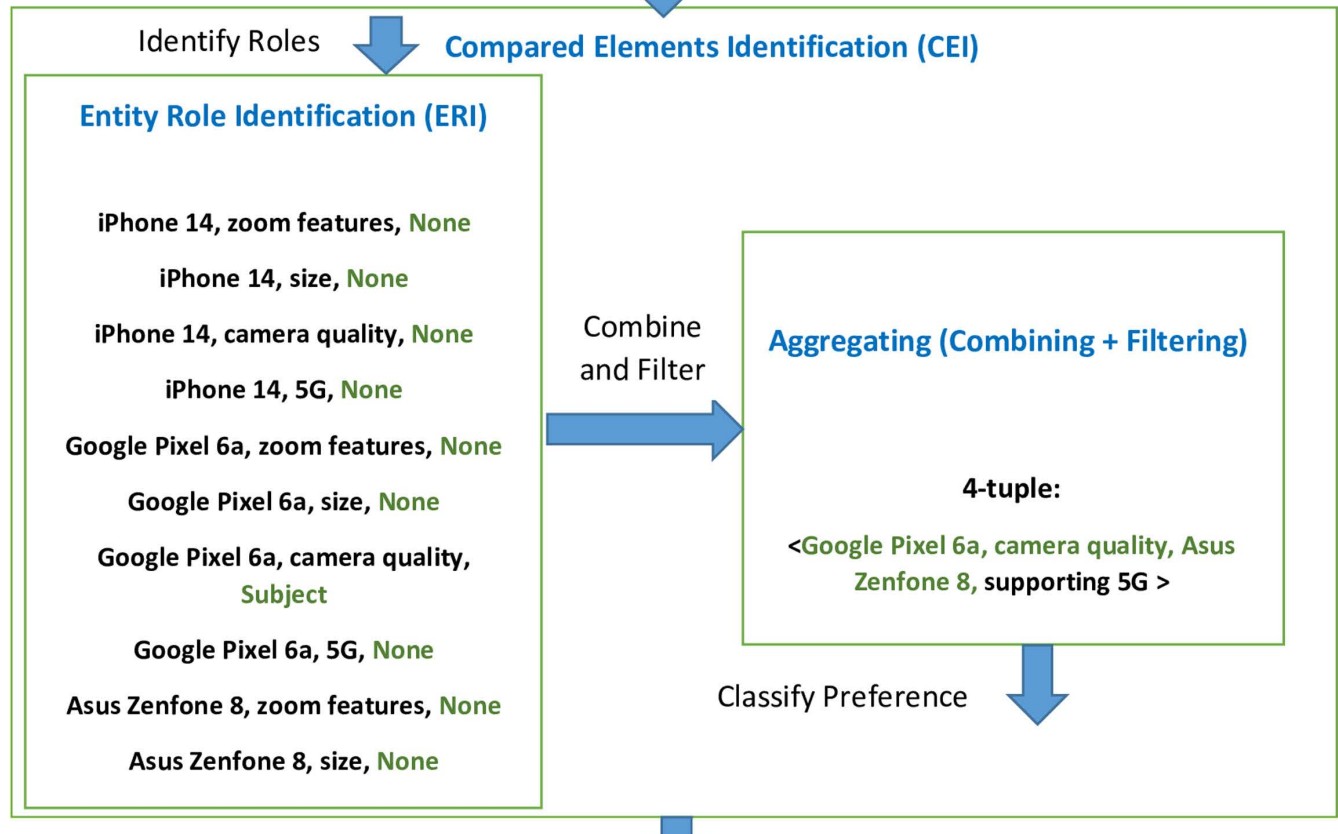

**Fig 2. Block Diagram for SCQRE.**

**Table 13. Case study comparisons across EE, CEI, CPC, and SCRE tasks.**

| Example | Task | Relation Analysis |
|---|---|---|
| Example 1 | Q | *Q1: Is iPhone A much more reliable than iPhone B but has a similar design?* |
| | EE | *Entities = [iPhone A, iPhone B]*<br>*Aspects = [reliable, design]*<br>*Constraints=[""]* |
| | CEI | *4-tuple1 = <iPhone A, iPhone B, reliable, "">*<br>*4-tuple2 = <iPhone A, iPhone B, design, "">* |
| | CPC | *(Q1, 4-tuple1)◊ Strong Better*<br>*(Q1, 4-tuple2)◊ Equal* |
| | SCRE | *CR1 = <iPhone A, iPhone B, reliable, "", Strong Better>*<br>*CR2 = <iPhone A, iPhone B, design, "", Equal>* |
| Example 2 | Q | *Q2: Is Samsung A better at the camera than Samsung B and Samsung C, but worse at battery life?* |
| | EE | *Entities = [Samsung A, Samsung B, Samsung C]*<br>*Aspects = [camera, battery life]*<br>*Constraints=[""]* |
| | CEI | *4-tuple1 = <Samsung A, [Samsung B, Samsung C], camera, "">*<br>*4-tuple2 = <Samsung A, [Samsung B, Samsung C], battery life, "">* |
| | CPC | *(Q2, 4-tuple1)◊ Better*<br>*(Q2, 4-tuple2)◊ Worse* |
| | SCRE | *CR1 = <Samsung A, Samsung B, camera, "", Better>*<br>*CR2 = <Samsung A, Samsung B, battery life, "", Worse>* |

In the realm of subjective comparative analysis, the interpretation of preferences and evaluations between entities is often influenced by personal opinions imbued with subtle emotional undertones. The GoEmotions dataset, which underpins the roberta-base-go_emotions model, provides a rich tapestry of emotional contexts, equipping the model with the necessary tools to adeptly navigate and interpret nuanced expressions. This capability is crucial for subjective comparative relation extraction tasks. By accurately identifying these emotional nuances, the model enhances the extraction of entities, aspects, and preferences—components where emotional contexts significantly impact user comparisons.

The fine-tuning of the roberta-base-go_emotions model on the GoEmotions dataset equips it to excel in scenarios where user preferences are inherently tied to emotional content. Its multi-label classification capability allows the model to recognize overlapping and co-occurring emotional states, providing a detailed representation of the input's sentiment. This added layer of precision enables the model to remain highly sensitive to emotional subtleties that are often overlooked by traditional NLP models. As a result, roberta-base-go_emotions is particularly well-suited for tasks involving subjective comparative questions, where the ability to detect and incorporate emotional nuances significantly enhances the accuracy, relevance, and contextual appropriateness of the extracted relations.

**3.3.2. Element Extraction (EE).** At the onset of our pipeline is the Element Extraction (EE) phase. This stage breaks down into three primary objectives: identifying Entities, Aspects, and Constraints within a question. Designed as a sequence tagging challenge, the BIO tagging scheme is employed. Given a token sequence from a question, the goal is to produce a corresponding BIO tag sequence.

To achieve this, the traditional BIO scheme is enhanced by adopting a specialized set of labels: {B-E, I-E, O-E, B-A, I-A, O-A, B-C, I-C, O-C}. These labels define whether a token is at the "Beginning", "Inside", or "Outside" of an "Entity", "Aspect", or "Constraint". An illustration of this scheme is provided in Table 14.

Rather than opting for separate classifiers that independently optimize Entity, Aspect, and Constraint Extraction, a Multi-Task Learning (MTL) approach is adopted. This strategy organizes a unified deep classification model, merging the tasks mentioned above. The MTL model's design for the EE task is displayed in Fig 3.

**Table 14. Illustration of BIO tags for different Extraction Tasks using a sample question.**

| Task/Question | Which Samsung smartphone has better quality than iPhone 12 with a price below 600 $? |
|---|---|
| Entity Extraction | O-E,O-E, O-E, O-E, O-E, O-E, O-E, B-E, I-E, O-E, O-E, O-E, O-E, O-E, O-E, O-E |
| Aspect extraction | O-A, O-A, O-A, O-A, O-A, B-A, O-A, O-A, O-A, O-A, O-A, B-A, O-A, O-A, O-A, O-A |
| Constraint extraction | O-C, B-C, I-C, O-C, O-C, O-C, O-C, O-C, O-C, B-C, I-C, I-C, I-C, I-C, I-C, O-C |

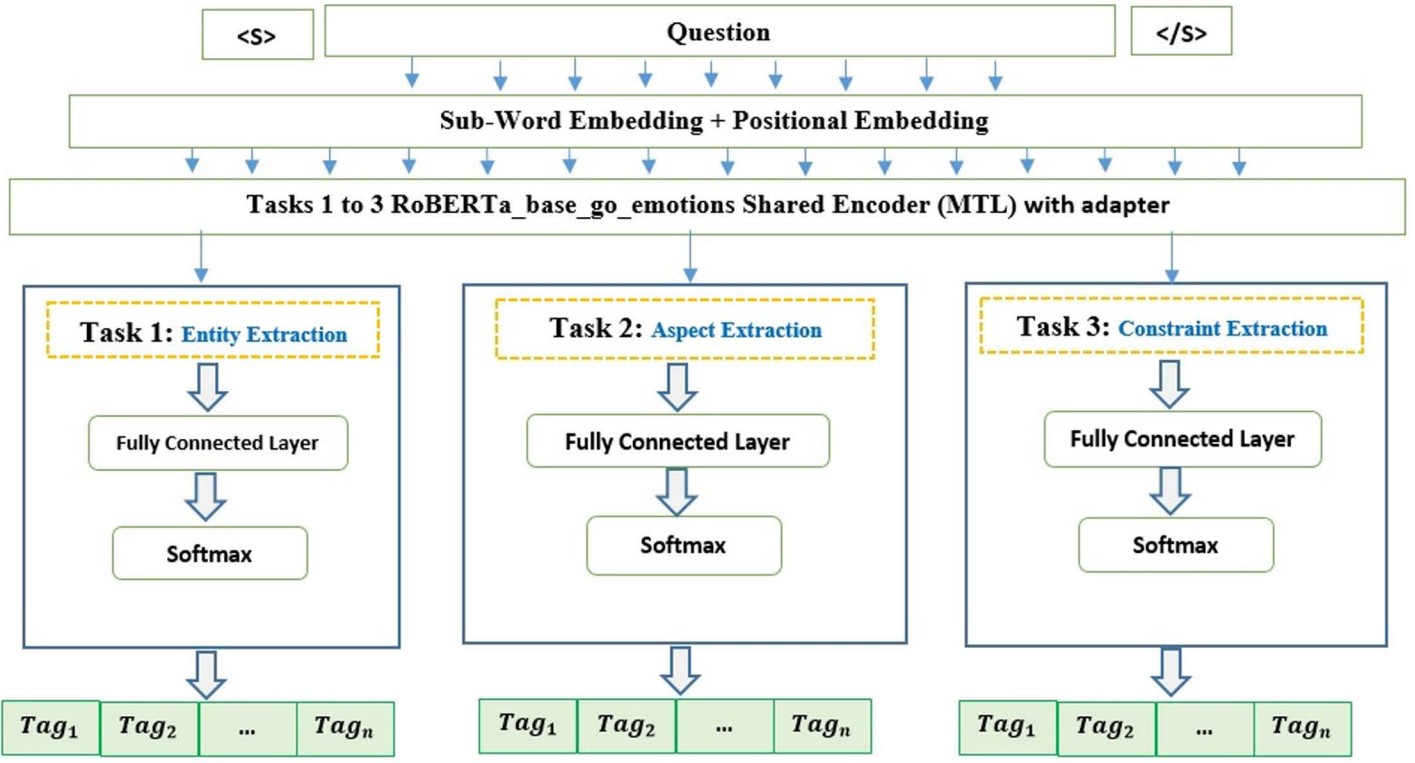

**Fig 3. Proposed MTL model for EE task.**

Leveraging the hard parameter-sharing strategy, as explored by Worsham and Kalita [33], we tap into the potential shared features among these relevant tasks. By sharing hidden layers and their parameters across all tasks and maintaining task-specific output layers, the risk of overfitting is reduced.

For the foundation of the classifier, we rely on the RoBERTa_base_go_emotions-based encoding layer [4], augmenting it with a fully connected layer. Additionally, we have integrated the "AdapterHub/roberta-base-pf-qnli" adapter, which has been pre-trained for question-answering tasks. This adapter enhances the model's ability to understand and respond to comparative questions, specifically focusing on identifying relationships between entities and aspects. By incorporating this adapter, our model gains a specialized capability to accurately interpret the nuances of subjective comparative queries without the need for extensive retraining.

### 3.3.3. Compared Elements Identification (CEI).

The Compared Elements Identification (CEI) stage is responsible for determining the roles of entities within a subjective comparative question and organizing them into a 4-tuple structure. This stage consists of two sub-processes: Entity Role Identification (ERI) and Aggregation.

3.3.3.1. Entity Role Identification (ERI); At the core of the CEI phase, the ERI process is instrumental in assigning roles to entities as "Subject", "Object", or "None". This classification is crucial for accurately distinguishing between subject and object entities, which cannot be solely achieved through BIO labeling used in the first phase for the following reasons:

1) **Variability in entity roles**: Entities within a single question may not always serve as subjects or objects in the comparative relation, necessitating a dedicated mechanism for role identification.

2) **Multiple comparative relations:** Subjective comparative questions often involve multiple comparative relations within a single context, requiring a method to differentiate entity roles across these relations.

3) **Entities without roles**: Some entities may not have subject/object roles but still relate to specific aspects, such as "quality display" or "RAM," as shown in Table 15, Case 8.

The ERI process leverages contextual cues and linguistic patterns specific to subjective comparative questions to ensure accurate role identification. In cases where entities are being compared for equality or using an XOR-type comparison (where the order doesn't matter), both entities are assigned the "Subject" role. This "Subject-Subject" classification reflects the interchangeable nature of the entities in these comparisons. This enhances the model's ability to navigate

**Table 15. Smartphone-SCQRE dataset samples utilizing NLI-M for CEI task.**

| # | Question Context | Auxiliary Sentence Construction | Label Assignment |
|---|---|---|---|
| 1 | *"Is the Samsung Galaxy M31 better than the Samsung Galaxy A50 in **software performance**?"* | "Samsung Galaxy M31 - software performance" | subject |
| | | "Samsung Galaxy A50 - software performance" | object |
| 2 | *"Does the Oppo F19 Pro have the same **sound quality** as the Realme X7?"* | "Oppo F19 Pro - sound quality" | subject |
| | | "Realme X7- sound quality" | subject |
| 3 | *"Which one is better for **taking pictures**: Google Pixel, Samsung Galaxy, or iPhone?"* | "Google Pixel - taking pictures" | subject |
| | | "Samsung Galaxy - taking pictures" | subject |
| | | "'iPhone - taking pictures" | subject |
| 4 | *"Is the Xiaomi Redmi Note 8 Pro the best compared to the Realme XT and Realme X2 Pro?"* | "Xiaomi Redmi Note 8 Pro – features" | subject |
| | | "Realme XT – features" | object |
| | | "Realme X2 Pro – features" | object |
| 5 | *"Is the Samsung Galaxy S7 better than the S6 regarding **RAM management** and **CPU performance**?"* | "Samsung Galaxy S7 - RAM management" | subject |
| | | "S6 - RAM management" | object |
| | | "Samsung Galaxy S7 - CPU performance" | subject |
| | | "S6 - CPU performance" | object |
| 6 | *"I want a smartphone. I like that the Samsung Galaxy A11 has a large **screen**, but I don't know if it will be as **reliable** as iPhone 8?"* | "Samsung Galaxy A11 – screen" | none |
| | | "iPhone 8 – screen" | none |
| | | "iPhone 8 – reliable" | subject |
| | | "Samsung Galaxy A11 – reliable" | subject |
| 7 | *"Does OnePlus 6 has been able to beat its ZenFone 5Z?"* | "OnePlus 6 – features" | subject |
| | | "ZenFone 5Z – features" | object |
| 8 | *"What are the best smartphones with **a built in stylus feature** with a good **quality display** and **RAM**, other than Samsung?"* | "Samsung - built in stylus feature" | none |
| | | "Samsung - quality display" | none |
| | | "Samsung – RAM" | none |

the intricacies of comparative relations and deliver more accurate and nuanced results. This classification establishes a framework for mapping the structure of comparative relations within the dataset, encompassing:

- **Input structuring for RoBERTa_base_go_emotions**: Inputs are methodically structured for RoBERTa_base_go_emotions by concatenating each question with auxiliary sentences, separated by RoBERTa_base_go_emotions's tokens ("<S>", "<\S>").

○ **Auxiliary sentence generation**: Inspired by Sun et al. [34], the NLI-M (Natural Language Inference with Multiple outputs) method is employed for its ability to handle varied comparative nuances, such as "Strong Better/Worse." This approach ensures clarity in classifying entities and is detailed in Appendix A.

○ **Generalization of aspects**: For questions without specified aspects, a generalized "features" label is applied (as in Table 15, Case 4 and 6).

○ **Classifying implicit preferences**: When relations in a question suggest preferences without explicitly stating them, these entities are classified as "subjects" (e.g., Table 15, Case 2, 3, and 6).

- **Adaptation of RoBERTa_base_go_emotions for ERI:** A fine-tuned RoBERTa_base_go_emotions model is used for sentence-pair classification to efficiently discern comparative roles.

- **Identification of compared aspects**: The ERI process also facilitates the identification of Compared Aspects, as the auxiliary sentences inherently link each entity's role to the corresponding aspect.

As shown in Fig 4, the deployment of the proposed RoBERTa_base_go_emotions-NLI-M model supports the ERI task by integrating these components into a unified architecture.

**3.3.4. Comparative Preference Classification (CPC).** In this section, we propose RoBERTa_base_go_emotions-NLI-SwapCPC, which integrates both the RoBERTa_base_go_emotions-NLI-M model and a data augmentation technique based on entity swapping and preference reversal for Comparative Preference Classification (CPC). This approach enhances the model's ability to generalize by leveraging both auxiliary sentence generation and data augmentation strategies to capture nuanced preferences.

The Comparative Preference Classification (CPC) phase addresses the challenge of sentence-pair classification, as illustrated in Fig 5. Using a refined version of the RoBERTa_base_go_emotions model, it processes two sentences and predicts the comparative preference of the question towards an auxiliary sentence. Given a question and its 4-tuple comparative relation (<Subject Entity, Object Entity, Compared Aspect, Constraint>), the task is to predict the preference label using the NLI-M method for generating auxiliary sentences. For example, for the question "*Why is the iPhone 10's camera inferior to the iPhone XS's?*" the auxiliary sentence "iPhone 10-camera versus iPhone XS-camera" generates the label "Worse."

We define 14 possible preference categories for subjective comparative questions, including Better, Strong Better, Worse, Strong Worse, Equal, XOR-Better, XOR-Strong Better, XOR-Equal, XOR-Worse, XOR-Strong Worse, X, X-Strong Better, X- Strong Worse, and Non-Gradable preferences. The output label falls into one of these categories. The model is fine-tuned for ten epochs using the Adam optimizer at a learning rate of 3e-5.

As illustrated in Fig 5, the RoBERTa_base_go_emotions-NLI-M model is deployed to process the input pairs and predict the comparative preference based on the auxiliary sentence.

**Defining question preference types in CPC;** In this section, we delve into the nuanced classification of question types that our Comparative Preference Classification (CPC) framework can distinguish. Our approach categorizes questions based on the clarity of preference direction and the definition of subject and object roles, providing a robust framework for analyzing the subtleties of comparative questions.

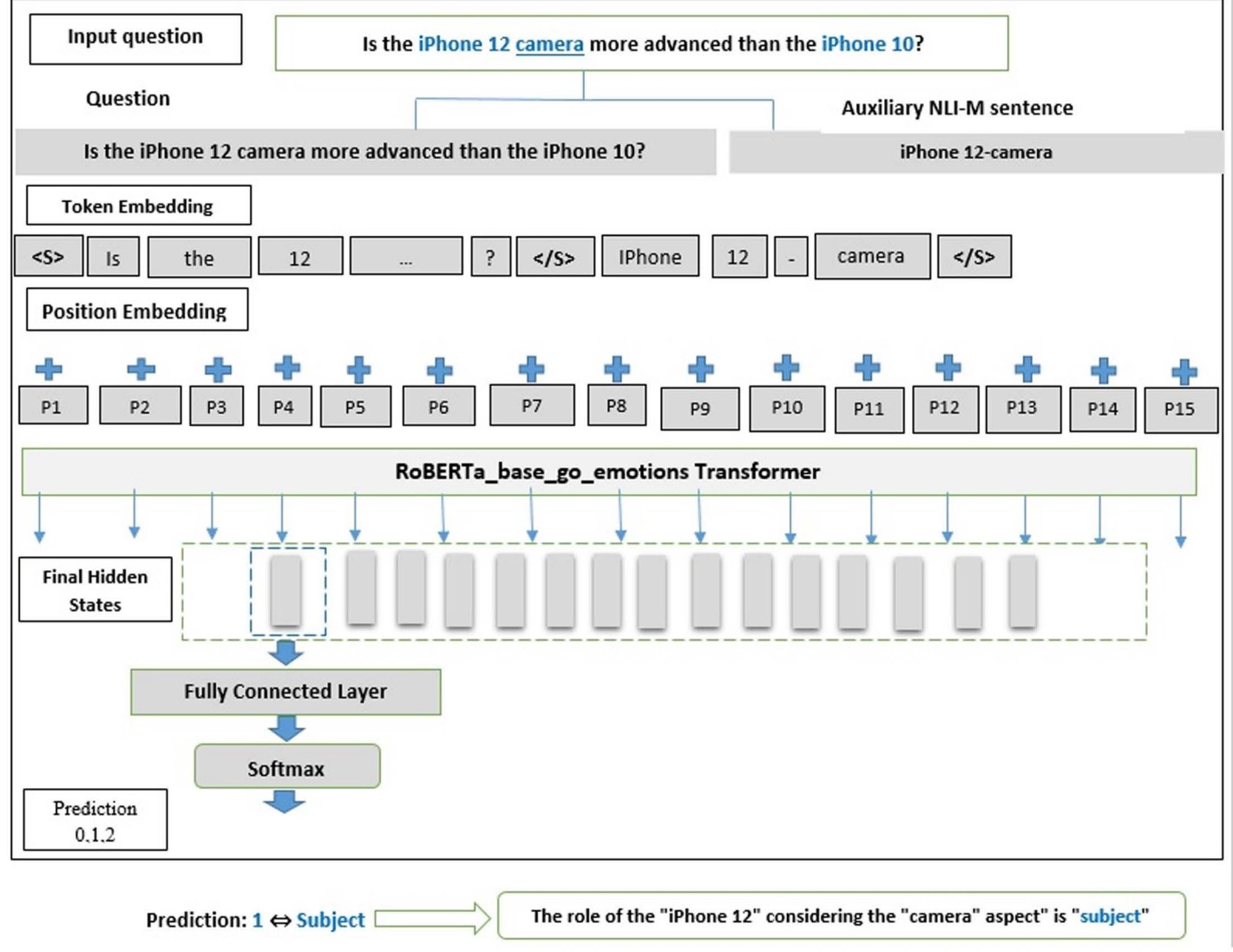

**Fig 4. Proposed RoBERTa_base_go_emotions-NLI-M Model Deployment for the ERI Task.**

- **Basic preferences**: Include straightforward comparisons where preference (Better/Worse/Equal) is clear, e.g., *"Does the iPhone 12 have better quality than the iPhone 9?"* (labeled as 'Better').

- **Preference intensity**: Reflects a significant degree of comparison, such as *"Why is Huawei's 5G technology much better than Nokia's?"* (labeled as 'Strong Better').

- **XOR-type preferences**: These questions specify a comparative direction (e.g., better, worse), but the roles of subject and object entities are left interchangeable. XOR-type questions allow flexible interpretations of which entity is preferred. For example:

  ◦ *"Which phone has better overall performance, Xiaomi Mi Max or Lenovo Phab 2 plus?"* (XOR-Better) – This question specifies one phone is better but doesn't fix which one is the subject or object.

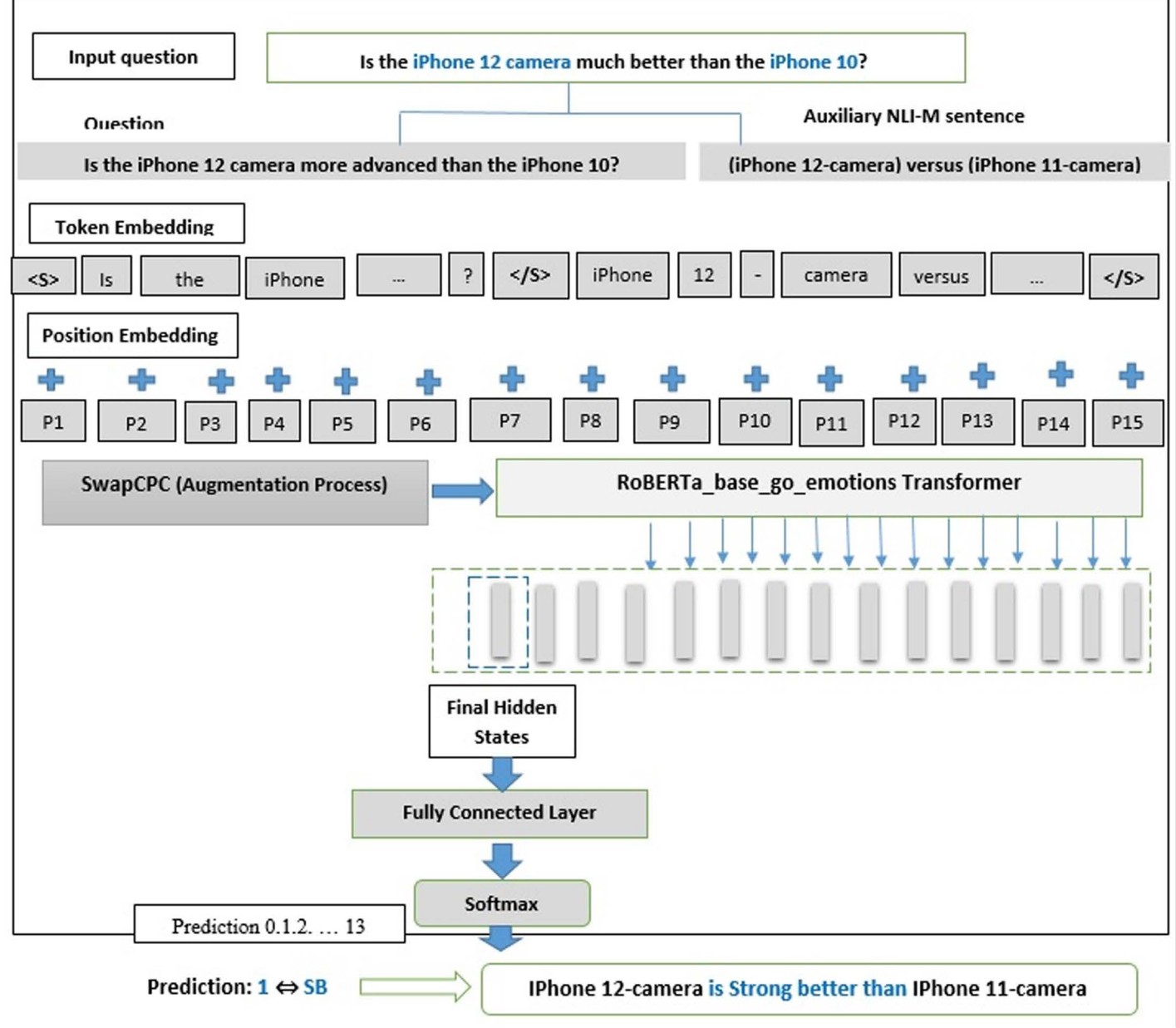

**Fig 5. Proposed RoBERTa_base_go_emotions-NLI-SwapCPC model deployment for the CPC task.**

- *"Samsung Galaxy A80 or iPhone 10, which one has lower quality?"* (XOR-Worse) – The query leaves the roles of subject and object undefined, allowing either phone to be considered the subject depending on context.

- **X-type preferences**: These questions have clearly identified subject and object entities, but the comparative direction is ambiguous. Often, they involve hypothetical or conditional comparisons. For example:

- *"How is the quality of the iPhone 11 compared to the Samsung Galaxy Note 10 Plus?"* (X) – The question invites analysis without concluding which phone is superior.

◦ *"How much better was the battery of the iPhone 13 mini compared to the iPhone 12 mini?"* (X-Strong Better) – This question explores potential battery improvement without explicitly stating a definitive preference.

• **Non-gradable preferences**: These comparisons emphasize distinct features without implying superiority or clear comparative roles. Non-Gradable questions focus on differences without ranking entities. Examples include:

• *"Is the Realme 8i in competition with Samsung A50?"* – This query focuses on market positioning without preference.

• *"Which iPhone series appearance looks distinguishable from the others?"* – This question highlights appearance uniqueness without ranking.

• *"What is the replacement phone for my Xiaomi Redmi Note 10 Lite among Oppo A54 and Vivo Y20? It must cost less than $1000."* – The question seeks a replacement based on price, without comparing performance or quality.

In classifying preferences, we account for both clear and incomplete preference types, including X-types, XOR-types, and Non-Gradable categories, which either leave the direction of preference or the roles of entities undefined. This flexible framework captures the variability in subjective comparative questions, providing structured analysis for each question type within the CPC framework.

**Data augmentation via entity swapping and reversing preference;** In addition to the standard comparative preference classification, we improved the model's learning capability through data augmentation, specifically using entity swapping and reversing preferences. For example, if the original question is, "*Is the iPhone 12 camera significantly more appealing than the iPhone 10?*" with iPhone 12 as the subject entity, iPhone 10 as the object entity, and camera as the compared aspect, the typical comparison would be:

• (iPhone 12-camera) versus (iPhone 10-camera), resulting in the label *Strong Better*.

To introduce more diversity and generalization into the model, we generate a reverse comparison:

• (iPhone 10-camera) versus (iPhone 12-camera), which would result in the label *Strong Worse*.

This swapping and reversing process exposes the model to both sides of the comparison, helping it to capture subtle shifts in preferences. It enhances the model's ability to generalize across different comparative structures, ensuring that it doesn't overfit to specific roles of subject and object entities, thus learning more robust patterns in comparative relations.

In our Comparative Preference Classification (CPC) task, we manage 14 distinct CPC labels that reflect various preference types between two entities. Most of these labels are included in our entity-swapping augmentation strategy, with sentence-pair classification applied to pair comparative questions with reversed comparative statements. However, labels such as *Non-Gradable*, *X*, *XorB*, *XorE*, *XorW*, *XorSB*, and *XorSW* are excluded from this approach because they signify non-preferential or ambiguous comparisons. Reversing these comparisons would not provide meaningful training data since their nature doesn't involve clear gradable preferences.

By incorporating this entity swapping and Reversing Preference technique, the model is exposed to a richer and more diverse set of comparative questions, ultimately improving its performance in the CPC task. Initial experiments show that augmenting the dataset using entity swapping results in a significant improvement in accuracy when handling XOR-type preferences, further supporting its efficacy.

## 4. Experimental results

In this section, the experimental setup is delineated. The subsequent sections detail the hyper-parameter tuning process and conclude with a comprehensive evaluation of the proposed model across various tasks, including EE, CEI, CPC, and SCRE.

**Table 16. Hyper-parameter setup for the proposed model across all tasks.**

| Parameters | EE | CEI | CPC |
|---|---|---|---|
| Batch Size | 8 | 64 | 64 |
| Max-Seq Length | 100 | 100 | 100 |
| Number of Epochs | 5 | 5 | 5 |
| Optimizer | AdamW | Adam | Adam |
| Learning Rate | 5e-5 | 3e-5 | 3e-5 |
| Dropout-Rate | 0.15 | 0.0 | 0.0 |

## 4.1. Hyper-parameter optimization

We conducted thorough experiments to determine optimal hyper-parameters for each component of the model—Element Extraction (EE), Compared Elements Identification (CEI), and Comparative Preference Classification (CPC). The selected hyper-parameter configurations, detailed in Table 16, were consistently employed for the EE, CEI, and CPC models without further changes.

## 4.2. Element Extraction (EE) evaluation

We assessed the Element Extraction (EE) task, encompassing three subtasks: Entity Extraction ($T_1$), Aspect Extraction ($T_2$), and Constraint Extraction ($T_3$). Each subtask is initially evaluated individually, followed by the evaluation of the multi-task learning (MTL) EE model and a subsequent comparison with other methods. Evaluations were performed using the Smartphone-SCQRE and Brands-CompSent-19-SCQRE dataset, employing stratified 5-fold cross-validation to ensure unbiased analysis. The results, which focus on Precision, Recall, and F-score metrics, are detailed in Table 17. This table shows comparisons between single-task learning (STL), MTL, and the adapter-enhanced MTL (MTL + Adapter) models. The results indicate that across all datasets, the MTL + Adapter model significantly outperforms both STL and MTL models, particularly excelling in handling complex subtasks such as Constraint Extraction.

Overall, the MTL + Adapter approach demonstrates consistent superiority across all tasks and datasets, outperforming both STL and MTL models in terms of Precision, Recall, and F-scores. By incorporating the adapter, the model can fine-tune task-specific nuances, making it particularly adept at handling the more challenging tasks, such as Entity Extraction, Aspect Extraction, and Constraint Extraction. While MTL provides substantial improvements over STL by leveraging shared representations across related tasks, the adapter further boosts performance by allowing for more focused adjustments, especially in complex subtasks. The greatest improvements are seen in the Smartphone-SCQRE dataset, where the adapter's impact is most pronounced, although notable gains are also observed in the Brands-CompSent-19-SCQRE dataset, demonstrating the model's versatility and effectiveness across different domains.

Moreover, Table 17 reveals that performance on each subtask offers insights into the complexity of the domains represented by each dataset. Smartphone-SCQRE was the easiest to process, likely because smartphones have more standardized and well-defined entities and aspects, making them easier to identify consistently. In contrast, the Brands-CompSent-19-SCQRE dataset posed additional challenges, particularly in Constraint Extraction, where more subjective and nuanced comparisons, such as brand reputation, come into play. The results suggest that Entity Extraction is generally more straightforward across domains, while Aspect and Constraint tasks are more context-dependent in Brands-CompSent-19-SCQRE, where ambiguities and variability require more tailored modeling approaches. This variability across datasets highlights the need for flexible models capable of adapting to the specific challenges of different domains.

The STL Aspect Extraction model is further validated using the SemEval 2014 dataset across the Laptop and Restaurant domains. The results, presented in Tables 18 and 19, indicate the model's state-of-the-art performance in these domains.

**Table 17. Performance analysis of element extraction across five runs on the Smartphone-SCQRE and Brands-CompSent-19-SCQRE Datasets.**

| Task | Approach | Smartphone-SCQRE | Brands-CompSent-19-SCQRE |
|---|---|---|---|
| **Entity Extraction** | STL | Precision: 0.930 | Precision: 0.875 |
| | | Recall: 0.917 | Recall: 0.890 |
| | | F-score: 0.924 | F-score: 0.883 |
| | MTL | Precision: 0.938 | Precision: 0.880 |
| | | Recall: 0.934 | Recall: 0.895 |
| | | F-score: 0.937 | F-score: 0.887 |
| | MTL + Adapter | Precision: 0.943 | Precision: 0.895 |
| | | Recall: 0.955 | Recall: 0.905 |
| | | F-score: 0.949 | F-score: 0.898 |
| **Aspect Extraction** | STL | Precision: 0.821 | Precision: 0.755 |
| | | Recall: 0.805 | Recall: 0.750 |
| | | F-score: 0.812 | F-score: 0.753 |
| | MTL | Precision: 0.835 | Precision: 0.820 |
| | | Recall: 0.934 | Recall: 0.855 |
| | | F-score: 0.876 | F-score: 0.838 |
| | MTL + Adapter | Precision: 0.888 | Precision: 0.840 |
| | | Recall: 0.897 | Recall: 0.845 |
| | | F-score: 0.892 | F-score: 0.843 |
| **Constraint Extraction** | STL | Precision: 0.764 | Precision: 0.720 |
| | | Recall: 0.747 | Recall: 0.690 |
| | | F-score: 0.724 | F-score: 0.705 |
| | MTL | Precision: 0.793 | Precision: 0.725 |
| | | Recall: 0.691 | Recall: 0.710 |
| | | F-score: 0.731 | F-score: 0.718 |
| | MTL + Adapter | Precision: 0.796 | Precision: 0.750 |
| | | Recall: 0.843 | Recall: 0.775 |
| | | F-score: 0.795 | F-score: 0.762 |

**Table 18. Performance of the STL Aspect Extraction model on the SemEval 2014 dataset.**

| Domain | Precision | Recall | F-score |
|---|---|---|---|
| Laptop | 0.86 | 0.88 | 0.87 |
| Restaurant | 0.87 | 0.89 | 0.88 |

Lastly, we briefly mention the methods compared:

- Augustyniak et al. [28] used GloVe embeddings with BiLSTM enhanced by CRF.

- Tran et al. [35] introduced stacked BiGRU-CRF for sentiment features.

- Da'u and Salim [36] implemented a multi-channel CNN using word and POS embeddings.

- Feng et al. [37] developed PECSMT: a model combining BERT, Contextual Summary, and a Transmission network.

- Wu et al. [38] presented DeepWMaxSAT, merging logic knowledge with deep learning for Aspect Extraction.

**Table 19. Comparison of the STL Aspect Extraction model with other leading models.**

| Reference | Restaurant | Laptop | |
|---|---|---|---|
| Augustyniak et al [28]. | 0.85 | 0.80 | **1** |
| Tran et al [35]. | 0.85 | 0.78 | **2** |
| Da'u and Salim [36] | 0.86 | 0.80 | **3** |
| Feng et al [37]. | 0.88 | 0.85 | **4** |
| Wu et al [38]. | 0.85 | 0.81 | **5** |
| Zschornack Rodrigues Saraiva et al [39]. | 0.87 | 0.81 | **6** |
| Chen and Qian [40] | 0.87 | 0.83 | **7** |
| Wei et al [41]. | 0.84 | 0.84 | **8** |
| Shi et al [42]. | 0.85 | 0.81 | **9** |
| Pour and Jalili [43] | 0.88 | 0.83 | **10** |
| Wang et al [44]. | 0.88 | 0.85 | **11** |
| Proposed STL Model | 0.88 | 0.87 | |

- Zschornack et al. [39] used a POS-AttWD-BLSTM-CRF architecture.

- Chen and Qian [40] focused on aspect and context term distributions, proposing SoftProto for word associations.

- Wei et al. [41] employed the DE-CNN method for aspect pre-extraction.

- Shi et al. [42] merged pre-trained embeddings with BiLSTM and CRF for aspect term extraction.

- Pour and Jalili [43] proposed CNN+WIN, focusing on specific aspect term windows.

- Wang et al. [44] improved sequence tagger performance with post-processing methods.

The results in Tables 18 and 19 demonstrate that our proposed STL Aspect Extraction model achieves state-of-the-art performance in both the Laptop and Restaurant domains. Specifically, the model outperforms most of the leading methods, including Augustyniak et al. [28], Chen and Qian [40], and others, achieving an F-score of 0.87 for the Laptop domain and 0.88 for the Restaurant domain. Notably, it matches the performance of Feng et al. [37] and Wang et al. [44] but surpasses them in precision and recall balance, highlighting its effectiveness in aspect extraction.

## 4.3. Compared Elements Identification (CEI)

CEI consists of three stages: generating auxiliary sentences from the question's Entities and Aspects, adapting Entity Role Identification (ERI) for sentence-pair classification, and refining the RoBERTa_base_go_emotions model. The process

**Table 20. Performance of the CEI model across five runs on the Smartphone-SCQRE and Brands-CompSent-19-SCQRE datasets.**

| | Smartphone-SCQRE | | | Brands-CompSent-19-SCQRE | | |
|---|---|---|---|---|---|---|
| K-fold | Precision | Recall | F-score | Precision | Recall | F-score |
| 1 | 0.815 | 0.805 | 0.810 | 0.798 | 0.790 | 0.794 |
| 2 | 0.820 | 0.808 | 0.814 | 0.803 | 0.792 | 0.797 |
| 3 | 0.810 | 0.815 | 0.812 | 0.795 | 0.798 | 0.796 |
| 4 | 0.818 | 0.813 | 0.816 | 0.801 | 0.796 | 0.798 |
| 5 | 0.814 | 0.809 | 0.811 | 0.799 | 0.793 | 0.796 |
| **Average** | **0.815** | **0.810** | **0.813** | **0.799** | **0.794** | **0.796** |

results in 4-tuple comparative sets for each question, assuming accuracy from the Element Extraction (EE) phase. Entity roles are categorized as Subject, Object, or None. To address class imbalance, the model applies stratified k-fold cross-validation and Focal Loss, improving accuracy for underrepresented classes.

Evaluation on the Smartphone-SCQRE and Brands-CompSent-19-SCQRE datasets achieved a micro-F-score of 0.813 and 0.796, respectively (Table 20). The model shows superior performance for the "Subject" class, particularly in multi-relation questions (Table 21).

The results in Tables 20 and 21 indicate that the CEI model performs consistently across the five runs for both datasets. On the Smartphone-SCQRE dataset, it achieved an average F-score of 0.813, while on the Brands-CompSent-19-SCQRE dataset, it obtained an F-score of 0.796. The model demonstrates a balanced performance between precision and recall, suggesting its robustness in handling different class distributions. In Table 21, the model shows higher F-scores for the "Subject" class (0.826 for Smartphone-SCQRE and 0.816 for Brands-CompSent-19-SCQRE) compared to "Object" (0.789 and 0.773) and "None" (0.758 and 0.744), highlighting its proficiency in identifying primary entities in comparative questions. This consistency across both overall and class-specific metrics reflects the model's effectiveness in comparative element identification.

Fig 6 compares the CEI model to two other models on the Smartphone-SCQRE, Brands-CompSent-19-SCQRE, and Camera-COQE datasets. Camera-COQE was chosen for its unique annotations. Performance ranks are: Proposed

**Table 21. Class-specific average F-Scores of the CEI model across five runs on Smartphone-SCQRE and Brands-CompSent-19-SCQRE.**

| K-fold | Smartphone-SCQRE | | | Brands-CompSent-19-SCQRE | | |
| --- | --- | --- | --- | --- | --- | --- |
| | Subject | Object | None | Subject | Object | None |
| 1 | 0.825 | 0.785 | 0.755 | 0.815 | 0.770 | 0.740 |
| 2 | 0.830 | 0.790 | 0.760 | 0.818 | 0.774 | 0.745 |
| 3 | 0.823 | 0.788 | 0.757 | 0.812 | 0.772 | 0.742 |
| 4 | 0.828 | 0.792 | 0.759 | 0.817 | 0.776 | 0.748 |
| 5 | 0.826 | 0.789 | 0.758 | 0.816 | 0.773 | 0.744 |
| **Average** | **0.826** | **0.789** | **0.758** | **0.816** | **0.773** | **0.744** |

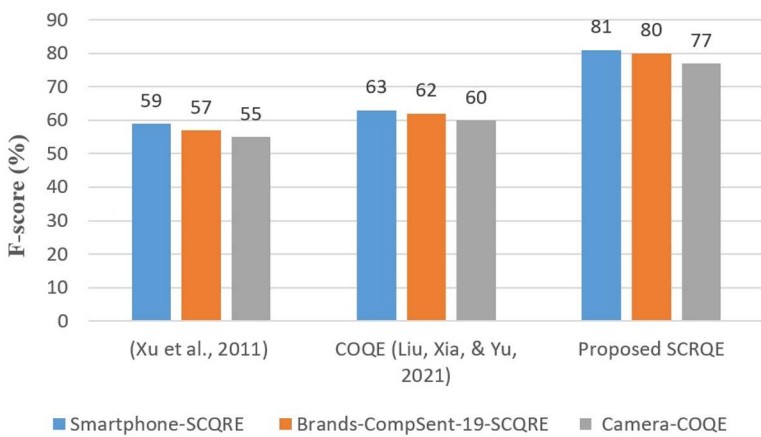

**Fig 6. Comparison of the CEI model with other studies on two datasets.**

CEI model> COQE> Xu et al. [16]. The proposed CEI model shows a significant improvement over both COQE and Xu et al. [16], achieving an F-score of 81 on Smartphone-SCQRE, 80 on Brands-CompSent-19-SCQRE, and 77 on Camera-COQE, demonstrating its adaptability and superior performance across different datasets.

#### 4.4. Comparative Preference Classification (CPC)

The CPC evaluation focuses on the effectiveness of the RoBERTa_base_go_emotions-NLI-SwapCPC model, assuming a successful preceding Compared Elements Identification (CEI). It treats the CPC as a sentence-pair classification, using an auxiliary sentence for input. The model's performance on the Smartphone-SCQRE and Brands-CompSent-19-SCQRE datasets is presented in Table 22, which shows the model achieving an impressive average F-score of 0.897 for Smartphone-SCQRE and 0.871 for Brands-CompSent-19-SCQRE.

Table 23 provides a detailed breakdown of the class-wise F-scores for the RoBERTa_base_go_emotions-NLI-SwapCPC model on both datasets over five runs, highlighting the model's ability to capture nuanced differences across various preference types. Specifically, the model demonstrates higher F-scores for ordinarily types compared to more

**Table 22. Performance Analysis of the RoBERTa_base_go_emotions-NLI-SwapCPC Model on Smartphone-SCQRE and Brands-CompSent-19-SCQRE Datasets across Five Runs.**

| K-fold | Smartphone-SCQRE | | | Brands-CompSent-19-SCQRE | | |
|---|---|---|---|---|---|---|
| | Precision | Recall | F-score | Precision | Recall | F-score |
| 1 | 0.925 | 0.910 | 0.895 | 0.918 | 0.905 | 0.890 |
| 2 | 0.927 | 0.913 | 0.896 | 0.920 | 0.907 | 0.892 |
| 3 | 0.926 | 0.915 | 0.897 | 0.919 | 0.908 | 0.891 |
| 4 | 0.928 | 0.917 | 0.899 | 0.921 | 0.910 | 0.893 |
| 5 | 0.926 | 0.914 | 0.896 | 0.920 | 0.909 | 0.892 |
| **Average** | **0.926** | **0.915** | **0.897** | **0.920** | **0.908** | **0.871** |

**Table 23. Class-wise F-scores of RoBERTa_base_go_emotions-NLI-SwapCPC model on Smartphone-SCQRE and Brands-CompSent-19-SCQRE datasets over five runs.**

| Dataset | K-fold | B | SB | W | SW | E | XorB | XorSB | XorE | XorW | XorSW | _X | SB_X | SW_X | Non-Grad | All |
|---|---|---|---|---|---|---|---|---|---|---|---|---|---|---|---|---|
| Smartphone-SCQRE | 1 | 0.920 | 0.918 | 0.915 | 0.914 | 0.913 | 0.905 | 0.903 | 0.901 | 0.899 | 0.897 | 0.890 | 0.888 | 0.886 | 0.870 | 0.897 |
| | 2 | 0.922 | 0.920 | 0.917 | 0.916 | 0.914 | 0.907 | 0.905 | 0.903 | 0.901 | 0.899 | 0.892 | 0.890 | 0.888 | 0.872 | 0.899 |
| | 3 | 0.921 | 0.919 | 0.916 | 0.915 | 0.913 | 0.906 | 0.904 | 0.902 | 0.900 | 0.898 | 0.891 | 0.889 | 0.887 | 0.871 | 0.898 |
| | 4 | 0.923 | 0.921 | 0.918 | 0.917 | 0.915 | 0.908 | 0.906 | 0.904 | 0.902 | 0.900 | 0.893 | 0.891 | 0.889 | 0.873 | 0.900 |
| | 5 | 0.922 | 0.920 | 0.917 | 0.916 | 0.914 | 0.907 | 0.905 | 0.903 | 0.901 | 0.899 | 0.892 | 0.890 | 0.888 | 0.872 | 0.899 |
| | Average | **0.922** | **0.920** | **0.917** | **0.916** | **0.914** | **0.907** | **0.905** | **0.903** | **0.901** | **0.899** | **0.892** | **0.890** | **0.888** | **0.872** | **0.897** |
| Brands-CompSent-19-SCQRE | 1 | 0.900 | 0.898 | 0.895 | 0.894 | 0.892 | 0.880 | 0.878 | 0.876 | 0.874 | 0.872 | 0.860 | 0.858 | 0.856 | 0.840 | 0.871 |
| | 2 | 0.902 | 0.900 | 0.897 | 0.896 | 0.894 | 0.882 | 0.880 | 0.878 | 0.876 | 0.874 | 0.862 | 0.860 | 0.858 | 0.842 | 0.873 |
| | 3 | 0.901 | 0.899 | 0.896 | 0.895 | 0.893 | 0.881 | 0.879 | 0.877 | 0.875 | 0.873 | 0.861 | 0.859 | 0.857 | 0.841 | 0.872 |
| | 4 | 0.903 | 0.901 | 0.898 | 0.897 | 0.895 | 0.883 | 0.881 | 0.879 | 0.877 | 0.875 | 0.863 | 0.861 | 0.859 | 0.843 | 0.874 |
| | 5 | 0.902 | 0.900 | 0.897 | 0.896 | 0.894 | 0.882 | 0.880 | 0.878 | 0.876 | 0.874 | 0.862 | 0.860 | 0.858 | 0.842 | 0.873 |
| | Average | **0.902** | **0.900** | **0.897** | **0.896** | **0.894** | **0.882** | **0.880** | **0.878** | **0.876** | **0.874** | **0.861** | **0.859** | **0.857** | **0.841** | **0.871** |

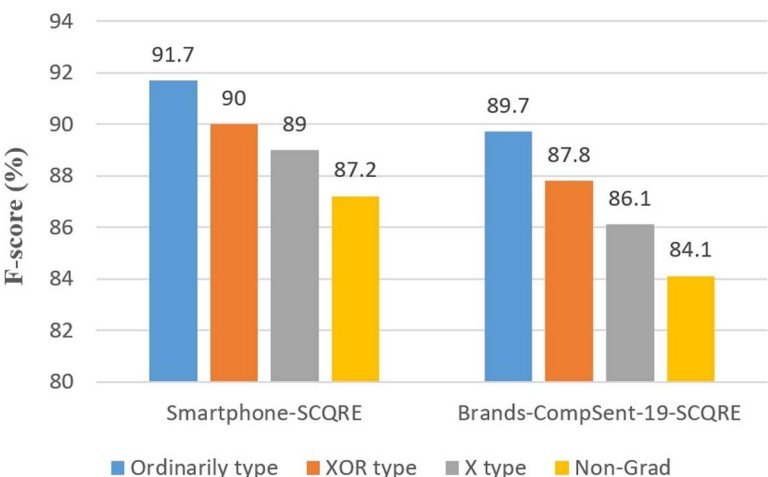

**Fig 7. F-scores for Coarse Preference Types using RoBERTa_base_go_emotions-NLI-SwapCPC on Smartphone-SCQRE and Brands-CompSent-19-SCQRE Datasets.**

complex categories such as XOR-types, X-types, and Non-Gradable preferences, indicating its robust performance in interpreting subjective comparative nuances.

To visualize these findings, Fig 7 depicts the F-scores for the coarse preference types presented in Table 23: ordinarily types, XOR types, X types, and non-gradable preferences. The ordinarily types include better, worse, strong better, and strong worse. XOR types consist of XOR-better, XOR-worse, XOR-strong better, and XOR-strong worse. X-types cover X, X-strong better, and X-strong worse. This figure illustrates the model's consistent proficiency with ordinarily types compared to the more complex categories.

We choose the CompSent-19 and Camera-COQE datasets for our comparative analysis despite their sentence-based focus because there are no closely related question-based datasets annotated specifically for Comparative Preference Classification (CPC). Given the lack of directly comparable datasets, these were selected due to their prevalent use in

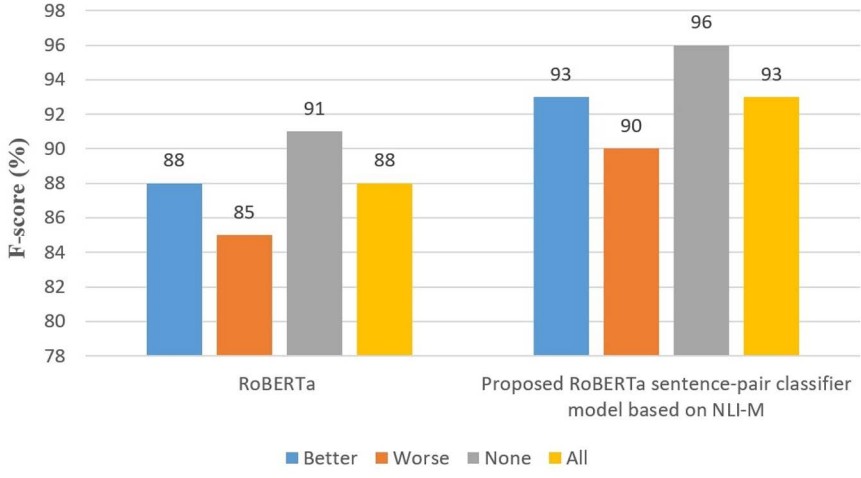

**Fig 8. RoBERTa_base_go_emotions-NLI-SwapCPC model vs. fine-tuned RoBERTa_base_go_emotions on CompSent-19 dataset.**

existing research, providing established benchmarks against which we can evaluate our model's performance. This selection allows us to contextualize our findings within the broader landscape of comparative analysis, offering a baseline for comparison against documented benchmarks in the field.

To evaluate the importance of the proposed CPC model (sentence-pair classification + SwapCPC) over single-sentence classification in comparative preference tasks, Fig 8 demonstrates a 5% improvement in performance. This improvement illustrates the added value of incorporating sentence-pair inputs for better context understanding in such tasks. A pivotal transformation from single-sentence to sentence-pair classification is realized through RoBERTa_base_go_emotions fine-tuning, enhanced by the SwapCPC technique. This transition is evidenced by a 5% performance increase in sentence-pair classification over single-sentence analysis on the CompSent-19 dataset in the domain of subjective comparative relation extraction, underscoring the enhanced capability of our approach to process and interpret subjective comparative nuances more effectively.

Furthermore, to evaluate our approach on other datasets and assess its generalization, we extended our evaluation to CompSent-19 and Camera-COQE, comparing our model's performance against other leading models in the field. Comparisons of the proposed CPC model against others on CompSent-19 and Camera-COQE datasets are shown in Table 24 and Fig 9. To align our model with the CompSent-19 and Camera-COQE benchmarks, several adjustments were made. First, we adapted our model's granular 'Strong Better'/'Strong Worse' categories to the binary 'Better'/'Worse' distinctions in these datasets, ensuring a coherent comparison. Second, we excluded question-specific categories like X-types and XOR-types,

**Table 24. RoBERTa_base_go_emotions-NLI-SwapCPC model vs. leading models on the CompSent-19 dataset.**

| Ref | Model | Better | Worse | None | All |
|---|---|---|---|---|---|
| Ganapathibhotla and Liu [45] | Rule-based Baseline | 0.65 | 0.44 | 0.90 | 0.82 |
| Panchenko et al [9]. | InferSent + XGBoost | 0.75 | 0.43 | 0.92 | 0.85 |
| Ma et al [24]. | ED-GAT BERT (Entity-aware Dependency-based Graph Attention Network) | 0.77 | 0.54 | 0.93 | 0.87 |
| Liu et al [8]. | SAECON (Sentiment Analysis Enhanced Comparative Network) | 0.78 | 0.56 | 0.93 | 0.87 |
| Li, et al [46]. | BERT-CLS | 0.77 | 0.57 | 0.91 | 0.855 |
| Proposed CPC model | RoBERTa_base_go_emotions-NLI-SwapCPC | 0.93 | 0.90 | 0.96 | 0.93 |

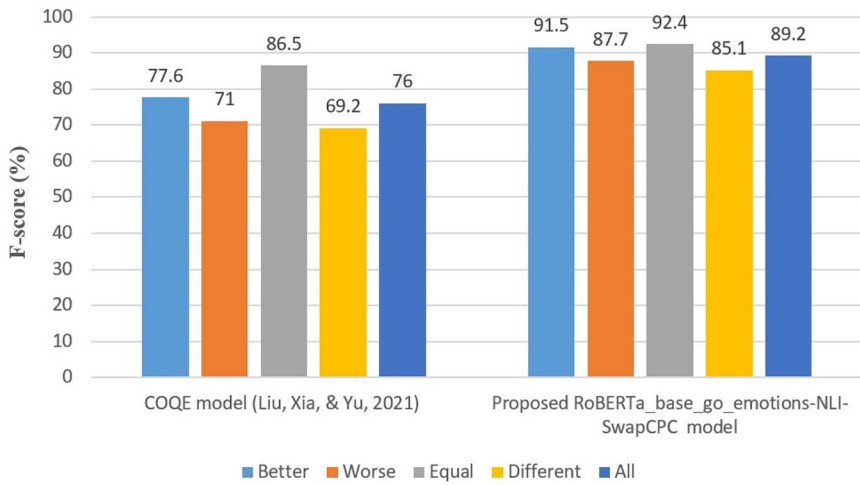

**Fig 9. RoBERTa_base_go_emotions-NLI-SwapCPC model vs. COQE model on Camera-COQE dataset [ 8].**

focusing on standard comparative types to ensure balanced evaluation. Lastly, we aligned the 'Different' category from Camera-COQE with our model's 'Non-Gradable' comparisons, acknowledging distinct qualities that resist simple linear comparison. These adjustments demonstrate our model's adaptability while maintaining consistency across diverse datasets.

Our comprehensive evaluation of the CPC aspect underlines the robustness and versatility of the RoBERTa_base_go_emotions-NLI-SwapCPC model in deciphering complex comparative preferences across multiple domains. This is further validated by the comparisons with leading models on the CompSent-19 and Camera-COQE datasets, as presented in Table 24 and Fig 9. The RoBERTa_base_go_emotions sentence-pair representation stands out, enhancing the model's ability to capture subtle comparative preferences. Table 24 highlights that our model achieves a higher F-score of 0.93 on CompSent-19, demonstrating superior performance when compared to other leading models. This evaluation not only validates the model's effectiveness in handling nuanced comparative queries but also showcases its ability to generalize across diverse datasets, paving the way for further explorations into enhancing its interpretative depth and accuracy.

## 4.5. Subjective Comparative Relation Extraction (SCRE)

This segment analyzes the performance of the SCRE model, examining it from multiple facets. Firstly, the model's effectiveness is viewed in terms of extracted comparative elements which are foundational to the construction of comparative relations. A tripartite evaluation, focusing on the subtasks - EE, CEI, and CPC - forms the initial testing strategy, leading to the computation of the micro-average F-score for SCRE. The model is also tested in a sequential, three-stage pipeline: EE to CEI to CPC. This method, while efficient, carries the risk of errors propagating through stages. Inspired by Liu et al. [8], evaluation metrics are designed around three matching strategies: Exact, Proportional, and Binary. Subsequently, a comparative study comparing the SCQRE model against other models strengthens its position in the field of comparative relation extraction.

**4.5.1 . Performance Metrics of SCQRE: Insights from Multiple Matching Strategies.** In order to evaluate the SCQRE, we follow the methodologies in [8], which require the computation of Precision, Recall, and F1-score for all quintuples. The formulas to obtain these metrics are as follows:

$$Precision = \frac{\#correct}{\#predict}, Recall = \frac{\#correct}{\#gold}, \text{F1score} = \frac{2 \times Precision \times Recall}{Precision + Recall} \tag{1}$$

Where:

- **#predict** is the number of predicted quintuple comparative relations (Subject Entity, Object Entity, Compared Aspect, Constraint, Comparative Preference).

- **#gold** is the number of quintuple comparative relations in the gold standard dataset.

- **#correct** is the count of correct quintuple comparative relations in the predictions.

Following the methodology from [8], we employ Precision, Recall, and F-score metrics using three matching strategies: Exact, Proportional, and Binary matching. An Exact match is a strict method where predicted comparative elements must exactly match the golden annotations to be considered correct. Binary and Proportional matches are more lenient. The Binary match checks if all elements of a predicted quintuple overlap with the golden annotation. The Proportional match computes the maximum overlap ratio between predicted and golden elements.

In assessing the accuracy of our model's predictions against the gold annotations, we originally described the process as a "metaphorical interpretation of intersection" to evaluate overlap in Binary and Proportional matches. However, to align with mathematical precision, we now clarify that the intersection is applied to sets of predicted and gold quintuples,

not individual elements as described in Xu et al. [47]. This approach provides mathematical rigor in quantifying extent of agreement between the model's predictions and the dataset annotations, accommodating the nuanced representation of comparative relations.

Specifically, the overlap, or #correct, is calculated by comparing the predicted and gold quintuples based on the three matching strategies. The number of correct quintuples is now defined as follows:

$$\#correct_E = \begin{cases} 1 & if\ p_i = g_i \\ 0 & otherwise \end{cases} \tag{2}$$

$$\#correct_B = \begin{cases} 1 & if\ p_i \bigcap g_i \neq \varnothing; \\ 0 & otherwise \end{cases} \tag{3}$$

$$\#correct_P = \begin{cases} 0 & if\ p_i \bigcap g_i = \varnothing \\ \frac{\sum_i len(p_i \bigcap g_i)}{\sum_i len(g_i)} & otherwise \end{cases} \tag{4}$$

Here, $p_i$ and $g_i$ represent the i-th predicted and gold quintuple, respectively, while len(·) indicates the length of the comparative element. The formulation ensures that the intersection is applied to the entire quintuples, improving the mathematical accuracy of the evaluation [47].

Fig 10 plots the performance of the model under these strategies, focusing on Precision, Recall, and F1-score based on five independent runs on the Smartphone-SCQRE dataset. Interestingly, the results suggest a higher recall than precision, indicating a tendency for negative-to-positive misclassifications. The Binary matching strategy proved to be the most permissive, followed by Proportional and Exact matching in terms of performance.

The separate sub-tasks and the overall SCRE task in a pipeline present contrasting evaluation results. The integrated approach has its limitations, primarily due to cumulative errors across stages.

**4.5.2. Comparing SCQRE Performance to Existing Models across Multiple Datasets.** In this section, the proposed SCQRE model's performance is evaluated in comparison to existing models across multiple datasets, including the Smartphone-SCQRE, Brands-CompSent-19-SCQRE, and Camera-COQE datasets. The evaluation focuses on several key tasks: Entity Extraction (EE), Compared Elements Identification (CEI), and Comparative Preference Classification (CPC). Table 25 provides a detailed breakdown of the SCQRE model's performance across different

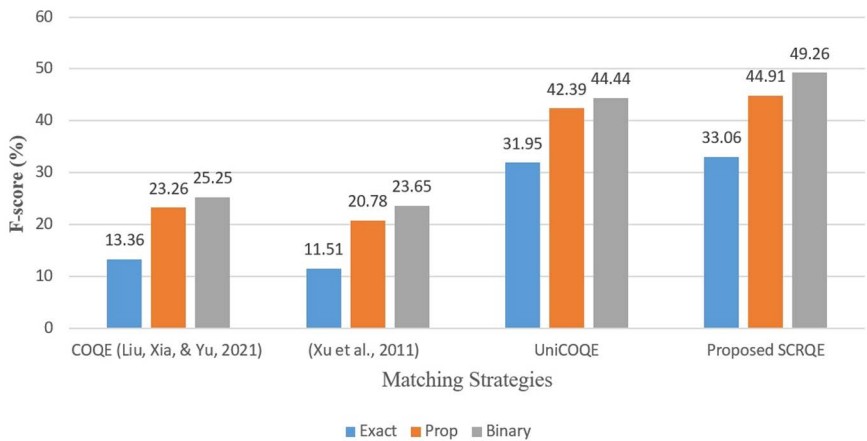

**Fig 10. Analysis of the SCQRE vs. Others on Camera-COQE Dataset [ 8] across Matching Strategies.**

**Table 25. Performance Metrics of the Proposed SCQRE on Smartphone-SCQRE Dataset over Five Runs.**

| Task | Smartphone-SCQRE | Brands-CompSent-19-SCQRE |
|---|---|---|
| **EE** | Precision: 0.943 | Precision: 0.900 |
| | Recall: 0.955 | Recall: 0.905 |
| | F-score: 0.949 | F-score: 0.902 |
| **CEI** | Precision: 0.866 | Precision: 0.830 |
| | Recall: 0.778 | Recall: 0.770 |
| | F-score: 0.802 | F-score: 0.798 |
| **CPC** | Precision: 0.852 | Precision: 0.820 |
| | Recall: 0.842 | Recall: 0.800 |
| | F-score: 0.837 | F-score: 0.810 |
| **Micro-Average F-score** | Precision: 0.887 | Precision: 0.850 |
| | Recall: 0.858 | Recall: 0.825 |
| | F-score: 0.863 | F-score: 0.836 |

**Table 26. F-scores of the SCQRE Model on the Smartphone-SCQRE and Brands-CompSent-19-SCQRE Datasets.**

| Matching Strategies | Smartphone-SCQRE | | | Brands-CompSent-19-SCQRE | | |
|---|---|---|---|---|---|---|
| | Precision | Recall | F-score | Precision | Recall | F-score |
| **Exact** | 28.5 | 33.0 | 30.64 | 27.3 | 31.1 | 29.07 |
| **Proportional** | 40.2 | 46.1 | 43.15 | 39.4 | 43.0 | 41.19 |
| **Binary** | 47.6 | 54.2 | 51.03 | 46.0 | 52.1 | 48.98 |

matching strategies—Exact, Proportional, and Binary—on both the Smartphone-SCQRE and Brands-CompSent-19-SCQRE datasets. In contrast, Table 26 and Fig 10 present a comparative analysis of the SCQRE model against other state-of-the-art models like COQE, Xu et al., and UniCOQE, highlighting its strong performance in Exact, Proportional, and Binary matching strategies. To compare with COQE, Xu et al., and UniCOQE in Table 26, and to align with our model and datasets, instead of opinion word extraction, we consider constraint extraction and re-ran those models.

The performance of the proposed SCQRE model is evaluated against existing models on various datasets, with a focus on tasks such as Entity Extraction (EE), Compared Elements Identification (CEI), and Comparative Preference Classification (CPC). The results are presented in Table 25, which highlights the model's effectiveness across both the Smartphone-SCQRE and Brands-CompSent-19-SCQRE datasets. For Entity Extraction, the SCQRE model demonstrates high precision, recall, and F-score values across both datasets, with the Smartphone-SCQRE dataset showing an F-score of 0.949 and the Brands-CompSent-19-SCQRE dataset yielding 0.902. This consistency underscores the model's robustness in accurately identifying entities in comparative questions across different datasets.

However, when evaluating the model's performance in Compared Elements Identification (CEI), we observe a slight drop in the F-scores compared to the EE task. On the Smartphone-SCQRE dataset, the model achieves an F-score of 0.802, while on the Brands-CompSent-19-SCQRE dataset, it scores 0.798. These results suggest that CEI, which involves identifying the relationship between subject and object in comparative sentences, is a more challenging task, leading to lower scores compared to entity extraction.

For the Comparative Preference Classification (CPC) task, the SCQRE model performs relatively well, with F-scores of 0.837 on the Smartphone-SCQRE dataset and 0.810 on the Brands-CompSent-19-SCQRE dataset. The performance across the two datasets remains consistent, indicating that the model is proficient in classifying preferences between comparative entities. The micro-average F-scores for both datasets also reflect the model's overall strong performance. For the Smartphone-SCQRE dataset, the micro-average F-score is 0.863, whereas the Brands-CompSent-19-SCQRE

**Table 27. Analysis of the SCQRE vs. Others on the Smartphone-SCQRE and Brands-CompSent-19-SCQRE Datasets across Matching Strategies.**

| | Smartphone-SCQRE | | | CompSent-19-SCQRE | | |
|---|---|---|---|---|---|---|
| **Matching Strategies** | **Exact** | **Prop** | **Binary** | **Exact** | **Prop** | **Binary** |
| **COQE (Liu, Xia, & Yu, 2021)** | 16.27 | 27.38 | 30.56 | 14.78 | 26.35 | 29.14 |
| **(Xu et al., 2011)** | 14.36 | 23.26 | 25.25 | 13.95 | 22.79 | 24.86 |
| **UniCOQE** | 28.39 | 40.11 | 48.91 | 26.87 | 39.55 | 46.06 |
| **Proposed SCRQE** | 30.64 | 43.15 | 51.03 | 29.07 | 41.19 | 48.98 |

dataset shows a slightly lower F-score of 0.836. These values highlight the model's generalization capability across different domains.

Table 26 delves into the model's performance across various matching strategies—Exact, Proportional, and Binary. The F-scores across these strategies vary, with the SCQRE model performing best under Binary matching with an F-score of 51.03 on the Smartphone-SCQRE dataset and 48.98 on the Brands-CompSent-19-SCQRE dataset. This shows that the model is highly effective in handling binary comparisons between entities, as binary matching offers a broader interpretation. In contrast, Exact matching yields lower F-scores of 30.64 on the Smartphone-SCQRE dataset and 29.07 on the Brands-CompSent-19-SCQRE dataset, suggesting that the model's performance is more challenged when strict matching criteria are required. Proportional matching, which allows for partial matches, provides a middle ground, yielding F-scores of 43.15 for the Smartphone-SCQRE dataset and 41.19 for the Brands-CompSent-19-SCQRE dataset.

The performance of the SCQRE model is further compared with other state-of-the-art models in Table 27. Compared to COQE (Liu, Xia, & Yu, 2021) and Xu et al. (2011), the SCQRE model consistently outperforms both across all matching strategies. For instance, in Exact matching, the SCQRE model achieves an F-score of 30.64 on the Smartphone-SCQRE dataset, while COQE achieves 16.27 and Xu et al. scores 14.36. A similar trend is observed for Binary matching, where the SCQRE model records an F-score of 51.03, significantly outperforming both COQE and Xu et al. In Proportional matching, the SCQRE model again leads with an F-score of 43.15 compared to COQE's 27.38 and Xu et al.'s 23.26. These results demonstrate that the SCQRE model offers superior precision and recall, making it a more robust solution for comparative quintuple extraction. To align with our model and datasets, instead of opinion word extraction, we considered constraint extraction and re-ran those models in Table 27.

The model's performance on the CompSent-19-SCQRE dataset follows a similar pattern, with the SCQRE model achieving higher F-scores across all matching strategies. Notably, the Proportional matching strategy results in an F-score of 41.19, further underscoring the model's ability to handle partially matching terms. In comparison, the COQE model achieves only 26.35, and Xu et al. scores 22.79, showcasing the superior performance of the SCQRE model in handling more complex comparisons. The Binary and Exact matching strategies similarly highlight the SCQRE model's superiority, with respective F-scores of 48.98 and 29.07.

Fig 10 evaluates the performance of the SCQRE model on the Camera-COQE dataset, where it is again compared to COQE and Xu et al. The SCQRE model achieves its highest F-scores across all matching strategies, including Exact (33.06), Proportional (44.91), and Binary (49.26). These results further confirm that the SCQRE model is more adept at handling comparative preference tasks compared to existing models, such as COQE and Xu et al., which achieve lower scores across all strategies. Additionally, the UniCOQE model, while competitive, falls short of matching the SCQRE model's performance, especially in Binary matching, where UniCOQE records an F-score of 44.44, compared to 49.26 for the SCQRE model. To align the Camera-COQE dataset and COQE, Xu et al., and UniCOQE models, we altered our model to extract opinion words instead of constraint extraction.

In summary, the proposed SCQRE model not only surpasses existing models in terms of F-scores but also demonstrates its versatility and generalization across multiple datasets and domains. The superior performance in tasks such

as Entity Extraction, Comparative Preference Classification, and matching strategies highlights the effectiveness of the SCQRE model in handling subjective comparative relations and extracting comparative quintuples. The results show that the SCQRE model is a state-of-the-art solution for comparative relation extraction, offering strong performance across the Smartphone-SCQRE, Brands-CompSent-19-SCQRE, and Camera-COQE datasets.

### 4.5.3. Comparison with prompt-based models.

To assess the effectiveness of the proposed model against recent language models, we conducted a series of experiments using **GPT-3.5-turbo-0613** [48], **Llama-2-70b-chat** [49], and **Qwen-1.5-7B-Chat** [50] models, all released in 2023. These models were evaluated across four primary tasks: Element Extraction (EE), Comparative Entity Identification (CEI), Comparative Preference Classification (CPC), and the overall quintuple extraction task, referred to as Subjective Comparative Relation Extraction (SCRE). We employed prompt engineering techniques to guide these models in generating accurate comparative relation quintuples.

While these generative models can indeed be fine-tuned with dedicated classification layers, we chose to focus on prompt-based evaluations to provide a straightforward zero-/few-shot comparison across different model families. Fine-tuning GPT-3.5-turbo, for example, necessitates commercial API access, while Llama-2-70B and Qwen-1.5-7B require significant computational resources to fine-tune locally. Consequently, exploring the fine-tuning route for these models is a promising direction for future work, but lies beyond the scope of this study.

Given the nature of prompt-based models, we explored three prompt configurations:

- **Short Prompt**: A concise prompt aimed at eliciting a focused response.

- **Medium Prompt**: A moderately detailed prompt designed to provide sufficient context.

- **Large Prompt**: An extensive prompt offering comprehensive context to ensure the model fully understands the task.

Each prompt configuration was tested across two settings:

- **One-Shot**: A single example provided within the prompt.

- **Few-Shot**: Fourteen examples provided (corresponding to the 14 CPC classes).

Due to space constraints, we present an example of the Medium Prompt with a One-Shot configuration below.
**Example of medium prompt with one-shot**
#Medium Prompt with One-Shot Template
"You are an NLP model trained to extract subjective comparative relation quintuples from questions."
prompt = (
'Extract Subjective Comparative Relation Quintuples from the following questions.\n'
'Each quintuple should include: Subject Entity, Object Entity, Compared Aspect, Constraint, and Comparative Preference.\n'
'Follow these rules for each component:\n'
'- Subject Entity: Specific product or brand, E.g., "iPhone 12".\n'
'- Object Entity: Specific product or brand, E.g., "Google Pixel 5". Use "none" if not mentioned.\n'
'- Compared Aspect: Specific feature or characteristic, E.g., "battery life". Use "All" if not mentioned.\n'
'- Constraint: Any conditions or limitations. Use "none" if not mentioned. E.g., "while gaming".\n'
'- Comparative Preference: Must be one of the following types: Better (B), Strong Better (SB), Worse (W), Strong Worse (SW), Equal (E), XOR-Better (XorB), XOR-Strong Better (XorSB), XOR-Equal (XorE), XOR-Worse (XorW), XOR-Strong Worse (XorSW), X (Unspecified comparison), X-Strong Better (X-Strong Better), X-Strong Worse (X-Strong Worse), Non-Gradable (NonGrad). These preferences indicate the direction and intensity of the comparison.\n'
'Here are some few-shot examples:\n'
'{\n'

**Table 28. Comparative performance of models across EE, CEI, and CPC tasks for various prompt types and few-shot configurations on the smartphone-SCQRE Dataset.**

| Model | Prompt Type | Few-Shot Configuration | EE | CEI | CPC |
|---|---|---|---|---|---|
| GPT-3.5-turbo-0613 | Short | One-Shot | 70% | 68% | 65% |
| | | Few-Shot (14) | 80% | 78% | 75% |
| | Medium | One-Shot | 75% | 73% | 70% |
| | | Few-Shot (14) | **85%** | **83%** | **80%** |
| | Large | One-Shot | 65% | 68% | 62% |
| | | Few-Shot (14) | 78% | 75% | 70% |
| Llama-2-70b-chat | Short | One-Shot | 65% | 62% | 57% |
| | | Few-Shot (14) | 72% | 70% | 65% |
| | Medium | One-Shot | 69% | 67% | 62% |
| | | Few-Shot (14) | **77%** | **75%** | **69%** |
| | Large | One-Shot | 67% | 65% | 59% |
| | | Few-Shot (14) | 75% | 72% | 67% |
| Qwen-1.5-7B-Chat | Short | One-Shot | 45% | 40% | 38% |
| | | Few-Shot (14) | 52% | 48% | 45% |
| | Medium | One-Shot | 48% | 43% | 42% |
| | | Few-Shot (14) | **55%** | **50%** | **48%** |
| | Large | One-Shot | 46% | 40% | 38% |
| | | Few-Shot (14) | 52% | 48% | 45% |
| **Proposed Model** | – | | **95%** | **80%** | **84%** |

"'Question 1": "Does the Xiaomi Mi 11 have a much better camera than the OnePlus Nord but a similar user interface?",\n'

"'Quintuple 1-1": {"Subject Entity": "Xiaomi Mi 11", "Object Entity": "OnePlus Nord", "Compared Aspect": "camera", "Constraint": "none", "Comparative Preference": "Strong Better"},\n'

"'Quintuple 1-2": {"Subject Entity": "Xiaomi Mi 11", "Object Entity": "OnePlus Nord", "Compared Aspect": "user interface", "Constraint": "none", "Comparative Preference": "Equal"},\n'

'}\n'

'Questions:\n'

)

Tables 28 and 29 summarize the results of these experiments. Table 28 compares the models' performance in Element Extraction (EE), Comparative Entity Identification (CEI), and Comparative Preference Classification (CPC) across different prompt types and few-shot configurations. Table 29 further analyzes the models' performance in Subjective Comparative Quintuple Extraction (SCQRE) using different matching strategies such as exact match, partial match, and fuzzy match.

As we can see from these tables, the best-performing configuration was the Medium Prompt combined with Few-Shot (14 examples), with GPT-3.5-turbo-0613 performing the best overall. Below, we discuss the results by considering the outputs of each of these models and analyzing the specific errors encountered by each. Here are the detailed comparisons:

1. **Element Extraction (EE) (Aspects, Entities, and Constraints)**

- **GPT-3.5-turbo-0613:**

  - **Aspects**: Performed best in aspect extraction, particularly when relations formed around compared aspects. Explicit aspects were extracted effectively, but nuanced or implicit aspects were sometimes missed.

**Table 29. Subjective comparative quintuple extraction performance across matching strategies by prompt type and few-shot configuration on the smartphone-SCQRE dataset.**

| Model | Prompt Type | Few-Shot Configuration | Exact Match (F-Score) | Proportional Match (F-Score) | Binary Match (F-Score) |
|---|---|---|---|---|---|
| GPT-3.5-turbo-0613 | Short | One-Shot | 15% | 22% | 29% |
| | | Few-Shot (14) | 22% | 29% | 35% |
| | Medium | One-Shot | 19% | 25% | 31% |
| | | Few-Shot (14) | **27%** | **32%** | **37%** |
| | Large | One-Shot | 17% | 23% | 30% |
| | | Few-Shot (14) | 25% | 31% | 36% |
| Llama-2-70b-chat | Short | One-Shot | 9% | 17% | 25% |
| | | Few-Shot (14) | 17% | 23% | 29% |
| | Medium | One-Shot | 14% | 21% | 27% |
| | | Few-Shot (14) | **22%** | **27%** | **32%** |
| | Large | One-Shot | 12% | 19% | 26% |
| | | Few-Shot (14) | 20% | 25% | 30% |
| Qwen-1.5-7B-Chat | Short | One-Shot | 5% | 12% | 19% |
| | | Few-Shot (14) | 12% | 19% | 25% |
| | Medium | One-Shot | 9% | 16% | 22% |
| | | Few-Shot (14) | **17%** | **22%** | **28%** |
| | Large | One-Shot | 7% | 14% | 21% |
| | | Few-Shot (14) | 15% | 21% | 27% |
| **Proposed Model** | – | | **31%** | **43%** | **51%** |

- ◦ **Entities**: Handled basic entity extraction well but occasionally confused subject-object roles in complex relations.

- ◦ **Constraints**: Struggled with complex or implicit constraints, especially when multi-layered conditions were involved.

- **Llama-2-70b-chat**

  - ◦ **Aspects**: Frequently over-extracted aspects, particularly in simple cases, where irrelevant details were pulled in.

  - ◦ **Entities**: Accurately identified entities but was prone to over-extraction in some cases.

  - ◦ **Constraints**: Performed better than GPT-3.5 in handling implicit constraints, managing multi-layered conditions effectively.

- **Qwen-1.5-7B-Chat**

  - ◦ **Aspects**: Often missed or incompletely extracted multi-word aspects, especially in complex multi-relation tasks.

  - ◦ **Entities**: Frequently missed or partially extracted entities in multi-relation settings.

  - ◦ **Constraints**: Simplified or missed constraints unless explicitly stated.

2. **Comparative Entity Identification (CEI)**

- **GPT-3.5-turbo-0613**: Handled simple entity identification well but sometimes confused subject-object roles in multi-relation tasks.

- **Llama-2-70b-chat**: Generally extracted entities accurately but occasionally over-identified irrelevant components.

- **Qwen-1.5-7B-Chat**: Frequently missed or extracted incomplete entity names, especially in complex multi-relation scenarios.

3. **Comparative Preference Classification (CPC)**

- **GPT-3.5-turbo-0613**: Performed best overall in handling preference intensity (e.g., "better" vs. "much better"). However, it struggled with non-gradable preferences and **X-types** (e.g., **X-Strong Better**, **X-Strong Worse**), often failing to capture these distinctions.
- **Llama-2-70b-chat**: Performed worse than GPT-3.5 in CPC, particularly with **X-types, XOR-types**, and **non-gradable preferences**, frequently misclassifying these preferences. It was slightly better than **Qwen-1.5-7B-Chat**.
- **Qwen-1.5-7B-Chat**: Frequently oversimplified preferences, reducing nuanced classifications to basic "better" or "worse" distinctions and struggled with complex preferences like **X-types** and **non-gradable preferences**.

4. **Overall Subjective Comparative Quintuple Extraction (SCQRE)**

- **GPT-3.5-turbo-0613**: Most effective overall, handling **aspect extraction** and **preference intensity** well. It struggled with non-gradable preferences and **X-types** but remained reliable in simpler quintuple extractions. Occasional role confusion occurred in multi-relation tasks.
- **Llama-2-70b-chat**: Handled complex relationships well but struggled with **X-types, XOR-types**, and **non-gradable preferences**, making it less effective than GPT-3.5 in nuanced preference extraction. It was slightly better than **Qwen-1.5-7B-Chat** in overall quintuple extraction.
- **Qwen-1.5-7B-Chat**: The least reliable in SCQRE, often missing or simplifying key quintuple components, especially in complex multi-relation tasks.

As we can see from Tables 28 and 29, our specialized model demonstrated superior performance in the quintuple extraction tasks, particularly in CPC, where it exhibited greater accuracy and consistency across different configurations. This superior performance can be attributed to several key factors:

- **Task-specific optimization**: The model was specifically optimized for quintuple extraction, allowing it to handle the unique challenges of Comparative Preference Classification (CPC) more effectively.
- **In-depth training**: Extensive training on a task-specific dataset enabled the model to capture the nuances of comparative questions more accurately.
- **Specialized architecture**: Our model's architecture is fine-tuned to address the intricacies of comparative relations and preferences, providing more precise classifications.
- **Consistency in performance**: Unlike the prompt-based models, which showed performance variations depending on prompt size and few-shot settings, our model consistently delivered accurate results across configurations.

This comparison highlights the robustness of our model in handling the complexities of comparative questions, particularly in accurately classifying comparative preferences, which is crucial for generating meaningful and contextually relevant responses.

## 5. Conclusion and future work

This study addresses a gap in Automated Question Answering (AQA) systems by focusing on subjective comparative questions, which have been less explored due to their complexity. We introduce the SCRQE model, a multistage deep

neural network designed to extract five key elements—Subject Entity, Object Entity, Compared Aspect, Constraint, and Comparative Preference—enhancing AQA systems' ability to handle comparative and subjective questions.

SCRQE is built using RoBERTa_base_go_emotions, combined with the integration of multi-task learning (MTL), Natural Language Inference (NLI), and the adapter mechanism, which significantly enhanced performance across key tasks, including Element Extraction (EE), Comparative Entity Identification (CEI), and Comparative Preference Classification (CPC). We validated our model on three datasets: Smartphone-SCQRE and Brands-CompSent-19-SCQRE, showing high inter-annotator reliability and demonstrating the robustness of SCRQE in handling comparative relations across different domains.

The experiments showed SCRQE outperforming prompt-based models like GPT-3.5-turbo-0613, Llama-2-70b-chat, and Qwen-1.5-7B-Chat, especially in dealing with non-gradable preferences and complex relations. Our model's optimization for CPC and EE tasks proved crucial in surpassing these benchmarks.

A key achievement of this research is the introduction of a comprehensive set of 14 preference types and the novel effort to extract constraints in comparative relations, improving the precision of subjective comparative analysis. Our experiments on external benchmarks, including Camera-COQE, CompSent-19, and SemEval 2014, confirmed the model's consistency and robustness.

While we focused on prompt-based evaluations in this work, exploring fine-tuning approaches for large generative models (e.g., GPT-3.5, Llama 2, and Qwen) is a promising future direction that may yield further improvements for complex comparative tasks.

Future work will focus on developing an end-to-end deep learning model that handles EE, CEI, and CPC simultaneously, reducing error propagation and increasing efficiency. Expanding the model across more product categories is a key priority to further advance comparative question answering systems.

## Supporting information

**S1 Appendix A. Auxiliary sentence generation algorithm.**
(DOCX)

**S2 Appendix B. Analysis of XOR and X-Type preferences in comparative questions.**
(DOCX)

## Author contributions

**Conceptualization:** Marzieh Babaali, Afsaneh Fatemi, Mohammad Ali Nematbakhsh.

**Data curation:** Marzieh Babaali.

**Formal analysis:** Afsaneh Fatemi, Mohammad Ali Nematbakhsh.

**Investigation:** Marzieh Babaali, Afsaneh Fatemi, Mohammad Ali Nematbakhsh.

**Methodology:** Marzieh Babaali, Afsaneh Fatemi, Mohammad Ali Nematbakhsh.

**Project administration:** Afsaneh Fatemi.

**Resources:** Marzieh Babaali.

**Software:** Marzieh Babaali.

**Supervision:** Afsaneh Fatemi, Mohammad Ali Nematbakhsh.

**Validation:** Marzieh Babaali, Afsaneh Fatemi, Mohammad Ali Nematbakhsh.

**Visualization:** Marzieh Babaali.

**Writing – original draft:** Marzieh Babaali.

**Writing – review & editing:** Marzieh Babaali, Afsaneh Fatemi, Mohammad Ali Nematbakhsh.

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
