## [Decision Letter · Decision Letter 0]

19 Feb 2024

PONE-D-23-39359SCRQE: Subjective Comparative Relation Quintuple Extraction from Questions in Product DomainPLOS ONE

Dear Dr. Fatemi,

Thank you for submitting your manuscript to PLOS ONE. After careful consideration, we feel that it has merit but does not fully meet PLOS ONE’s publication criteria as it currently stands. Therefore, we invite you to submit a revised version of the manuscript that addresses the points raised during the review process.

**Please note that we have only been able to secure a single reviewer to assess your manuscript. We are issuing a decision on your manuscript at this point to prevent further delays in the evaluation of your manuscript. Please be aware that the editor who handles your revised manuscript might find it necessary to invite additional reviewers to assess this work once the revised manuscript is submitted. However, we will aim to proceed on the basis of this single review if possible.** **The reviewer has raised a range of concerns, including methodological clarity, the context and overall justification for this work compared to other work in the field, and appropriate comparisons with state-of-the-art reports on similar methods. Please ensure you address each of the reviewers' comments when revising your manuscript.**

We look forward to receiving your revised manuscript.

Kind regards,

Hugh Cowley

Staff Editor

PLOS ONE

Journal Requirements:

2. You indicated that ethical approval was not necessary for your study. We understand that the framework for ethical oversight requirements for studies of this type may differ depending on the setting and we would appreciate some further clarification regarding your research. Could you please provide further details on why your study is exempt from the need for approval and confirmation from your institutional review board or research ethics committee (e.g., in the form of a letter or email correspondence) that ethics review was not necessary for this study? Please include a copy of the correspondence as an "Other" file.

3. Please provide additional details regarding participant consent. In the ethics statement in the Methods and online submission information, please ensure that you have specified (a) whether consent was informed and (b) what type you obtained (for instance, written or verbal, and if verbal, how it was documented and witnessed). If your study included minors, state whether you obtained consent from parents or guardians. If the need for consent was waived by the ethics committee, please include this information.

4. In your Methods section, please include additional information about your dataset and ensure that you have included a statement specifying whether the collection and analysis method complied with the terms and conditions for the source of the data.

5. Please note that PLOS ONE has specific guidelines on code sharing for submissions in which author-generated code underpins the findings in the manuscript. In these cases, all author-generated code must be made available without restrictions upon publication of the work. Please review our guidelines at https://journals.plos.org/plosone/s/materials-and-software-sharing#loc-sharing-code and ensure that your code is shared in a way that follows best practice and facilitates reproducibility and reuse.

6. We note that your Data Availability Statement is currently as follows: All relevant data are within the manuscript and its Supporting Information files.

Reviewers' comments:

Reviewer's Responses to Questions

**Comments to the Author**

1. Is the manuscript technically sound, and do the data support the conclusions?

Reviewer #1: Partly

2. Has the statistical analysis been performed appropriately and rigorously? 

Reviewer #1: No

3. Have the authors made all data underlying the findings in their manuscript fully available?

Reviewer #1: Yes

4. Is the manuscript presented in an intelligible fashion and written in standard English?

Reviewer #1: Yes

5. Review Comments to the Author

Reviewer #1: The authors present a new dataset, denoted by the acronym SCRQD, containing

1,275 subjective comparative questions from the smartphone domain. Inspired

by (Liu et al, 2021), the questions are translated into (sets of)

quintuples denoting subject entity, object entity, compared aspect,

constraint, and preference. Together with the SCRQD dataset, a new SCRQE

system for extracting these quintuples from the input questions is

presented and evaluated.

The SCRQE system is designed as a pipeline with three stages that are

evaluated separately as well as in combination. The performance is partly

compared to other similar approaches although the details of the comparison

are not always clear.

A major drawback of the text lies in exaggerative claims about the

uniqueness of the presented approach which then result in the fuzzy

comparison. The authors claim that "there is no publicly available dataset

to derive subjective comparative relations from the questions". This is

definitely not true as comparatives in NLP are studied for decades and

simple search reveals tens of works in this regard.

Moreover, the presented approach is not specific to questions at all

despite the authors stating the opposite. They do not link or offer linkage

to possible answers and the RoBERTa-based analysis does not use anything

different from standard comparative sentence analysis. Other (mostly)

ignored sources of comparison are numerous aspect identification datasets

and approaches, although some are briefly mentioned in the evaluation of

the first stage of the system.

The Related works section offers relatively good overview of the approaches

(although concentrating on methods from 2021 and older) but these

approaches are not compared to the proposed system and the dataset

and no explanation of possible differences is offered.

The decisions for specific annotations and labels are in some cases unclear

even with the provided definitions and examples. For example, why

"reasonable" (line 275) is Non-Gradable? Is it different from

"good/better"? The definitions of preference types (lines 532-580) are

insufficiently explained. What is "a comparison without a clear reference

point" (XOR) or "a degree of uncertainty" (X)? For example, what are the

differences in meaning between the following four different annotations

from the text:

Is it better or not to upgrade A to B? (X)

What is the satisfaction level of A compared to B? (X-Strong Better)

Which phone is better overall performance, A or B? (XOR-Better)

Does A have better quality than B? (Better)

What are the inter-annotator agreement values for these fuzzy categories?

From the stylistic point of view, the text is flooded with too many

acronyms, making the comprehension and navigation in some parts quite

difficult.

Another example of "exaggeration" can be seen in the emphasis on the

Multi-Task Learning (MTL) approach used in the system where the actual

difference is just between multiple binary classifiers (here single task

approaches) and one multi-class classifier. Multi-class token

classification is the most standard way of named entity recognition which

exactly corresponds to the underlying task of the presented system. Here

using three separate single-task or binary classifiers would be more

unusual.

The schema in Figure 2 of the model for the entity extraction task is

misleading, displaying three separate label outputs from the input

question. The correct form is a sequence-to-sequence process resulting in

token labels (one to one for each token in the question).

The description of the second phase is unnecessarily long (lines 364-500,

more than 5 pages). The text mostly repeats the same (relatively simple)

procedure several times. For example, Tables 8 and (later) 11 are not

justified as the other 3 methods are not used at all here. Algorithm 1 does

not offer anything new to the text. One page should be enough here.

What is left unexplained, is the distinction between Subject and Object

(Table 9) where in some sentences there are only multiple subjects and in

others subjects with objects without explanation.

Is the second phase actually needed? What if the differentiation between

subject and object was expressed by the BIO labels in the first phase?

Dropout of 0.0 in Table 20 is slightly unusual. Moreover, this parameter is

not included in the list in 4.2. Hyper-Parameter Optimization with other

values.

Table 24 with a Comparison of Aspect Extraction models ignores some works

with better results, e.g.

Wu, Z., & Ong, D. C. (2021). Context-guided bert for targeted

aspect-based sentiment analysis. In Proceedings of the AAAI conference

on artificial intelligence (Vol. 35, No. 16, pp. 14094-14102).

or even the work [24] from the text presenting BERT-pair which was,

however, not included in this comparison table.

The details of the comparison evaluation are missing. Did you run the

compared models yourself? What is e.g. the "classification over

single-sentence on the CompSent-19 dataset" compared in Figure 7? What is

the Different label in Figure 8?

Table 26 displays relatively low values for the None class accompanied by

high numbers for the other two classes. What are the misclassifications

then? A detailed confusion matrix here could help to clarify.

Errors in text:

line 50, "we fine-tuned the SCRQD dataset" -> dataset is not fine-tuned

line 54, "superiority over existing model" -> "models"

line 156, "REDDY and KMahesh" -> "Reddy and Mahesh Babu"

line 162, "et al. [11]" -> "Liu et al. [11]"

Table 6, "(Q2, 4-tuple2)" in Example 1 and "(Q1, 4-tuple1)" in Example 2

incorrectly refer to the other question

Table 7. with Illustration of BIO tags is completely mistaken leading to

entity "then iPhone", aspects "better" and "a", and constraints "Samsung

smartphone" and "12 with a price below 600".

line 382, "As highlighted by Sun et al. [23]" - [23] is Devlin et al.

line 576, "questions that assessing the intensity or degree" -> "that assess"

line 769, "p_i intersection g_i" - not all quintuple elements are sets, are they?

line 779, "The Binary matching strategy emerged as the most efficient"

- claiming that the most permissive evaluation is "efficient" is strange

6. PLOS authors have the option to publish the peer review history of their article (what does this mean? ). If published, this will include your full peer review and any attached files.

**Do you want your identity to be public for this peer review?** For information about this choice, including consent withdrawal, please see our Privacy Policy .

Reviewer #1: No

---

## [Author Response · Author response to Decision Letter 1]

15 Apr 2024

Dear Editor,

We extend our deepest appreciation for the thorough evaluation of our manuscript and the opportunity to refine and resubmit it for consideration in PLOS ONE. The insightful comments and suggestions from the review process are invaluable, and we are confident that addressing them will substantially elevate the impact and clarity of our research.

Upon meticulous examination of the feedback provided, we have implemented comprehensive revisions to our manuscript. These modifications focus on enhancing methodological clarity, providing a more robust context and justification for our work, and making pertinent comparisons with the latest developments in our field, as advised.

We have responded to each point raised by the editor and reviewers in a detailed manner. These responses are documented in a Rebuttal Letter, which systematically addresses each comment and explains the modifications made to the manuscript. This document will be uploaded as a separate file under the label 'Response to Reviewers'.

Submission Documents:

Rebuttal Letter: A meticulously crafted document addressing each comment made by the academic editor and reviewers, uploaded as 'Response to Reviewers'.

Marked-Up Manuscript: This version displays all changes made to the original manuscript, clearly marked to facilitate review, submitted as 'Revised Manuscript with Track Changes'.

Unmarked Manuscript: A clean version of the revised manuscript, without tracked changes, labeled 'Manuscript'.

In alignment with your submission instructions, we are prepared to submit the revised manuscript by the stipulated deadline, along with the necessary documents. We will also re-examine our financial disclosure statement and update it as necessary in our cover letter.

Regarding the deposit of our laboratory protocols on protocols.io, we recognize the importance of this practice in enhancing research reproducibility and transparency. Accordingly, we have decided to deposit our protocols as recommended, contributing to the scientific community's efforts to enhance research integrity and accessibility.

We are eager to advance our contribution to the PLOS ONE community and are grateful for the guidance and support provided by your esteemed journal throughout this process. Should you need any further information or clarification, please feel free to reach out to us.

Warm regards,

Dr. Afsaneh Fatemi

---

## [Decision Letter · Decision Letter 1]

11 Jun 2024

PONE-D-23-39359R1SCRQE: Subjective Comparative Relation Quintuple Extraction from Questions in Product DomainPLOS ONE

Dear Dr. Fatemi,

Thank you for submitting your manuscript to PLOS ONE. After careful consideration, we feel that it has merit but does not fully meet PLOS ONE’s publication criteria as it currently stands. Therefore, we invite you to submit a revised version of the manuscript that addresses the points raised during the review process.

We look forward to receiving your revised manuscript.

Kind regards,

Leona Cilar Budler

Academic Editor

PLOS ONE

Additional Editor Comments:

There are some suggestions and comments from reviewers to improve your paper. Please take them into account to improve your paper.

Reviewers' comments:

Reviewer's Responses to Questions

**Comments to the Author**

1. If the authors have adequately addressed your comments raised in a previous round of review and you feel that this manuscript is now acceptable for publication, you may indicate that here to bypass the “Comments to the Author” section, enter your conflict of interest statement in the “Confidential to Editor” section, and submit your "Accept" recommendation.

Reviewer #1: (No Response)

Reviewer #2: All comments have been addressed

2. Is the manuscript technically sound, and do the data support the conclusions?

Reviewer #1: Partly

Reviewer #2: Yes

3. Has the statistical analysis been performed appropriately and rigorously? 

Reviewer #1: N/A

Reviewer #2: N/A

4. Have the authors made all data underlying the findings in their manuscript fully available?

Reviewer #1: Yes

Reviewer #2: Yes

5. Is the manuscript presented in an intelligible fashion and written in standard English?

Reviewer #1: Yes

Reviewer #2: Yes

6. Review Comments to the Author

Reviewer #1: The authors have provided a substantial revision of the original text, adding

a number of previously missing details of the work. While the new text has

significantly improved the critical points of the authors' work, several

new questions arise accompanying the added details.

One of the crucial points lies in the subject difference to most existing

datasets, i.e. to working with comparative questions instead of comparative

sentences. The authors point at the differences (and complications) between

these two data approaches, but a motivation for processing questions is

missing. A possible reason could be connected with "answers", that are

natural companions of questions, but these are not touched at all in the

text. The problems can be seen also in the dataset preparation procedure

where the authors admit that "due to the scarcity of questions that exactly

matched our research criteria, we modified existing questions and

formulated new ones to diversify our dataset" (line 511).

This motivation also concerns the two main data format differences to

previous approaches, i.e. the subject-object distinction (or the CEI task)

and the detailed comparison classification (or the CPC task) to 14 categories

instead of just Better/Worse. The impact of these changes is not evaluated

in the text. The dataset statistics of the subject-object (in Reply to

reviewer, missing in text) reveal that the number of non-standard cases,

i.e. different from subject-subject, is less than 10%. The examples of the

detailed comparison categories are not always clear, e.g. why in "Why is

the iPhone 10 camera inferior compared to the iPhone XS" denotes "inferior"

a "Strong Worse" category (line 942)?

In line 272, the presented SCQRD dataset is said to enable "robust model

training and evaluations across different languages and domains". How? The

dataset is limited to English and to one domain.

One of the main novelties of the presented approach lies in exploitation of

the RoBERTa base model as published by Facebook research in 2019. Since

that time, a number of newer, larger and more capable models have been

published. It thus may be expected that simply interchanging the underlying

pretrained model can substantially improve the results. Have you

experimented with newer models?

The only results which offer true comparison with related works and

a previously published dataset are the numbers in the chart of Fig 12

"Analysis of the SCRE vs. others on Camera-COQE [5] across matching

strategies". The authors still do not present other works that have

published better results on the Camera-COQE dataset, e.g.

Yang, Zinong, et al. "UniCOQE: Unified comparative opinion quintuple

extraction as a set." Findings of the Association for Computational

Linguistics: ACL 2023. 2023.

with 31.95 of Exact F1 score vs 23.8 by the current approach in Fig 12.

Fig 10 and 11 both present analysis results on the SCQRD dataset, first

with detailed values for the SCRQE system, then comparing the results to

two other approaches. However, the numbers for SCRQE do not match in the

figures, e.g. the Binary values are 43.12, 60.76 and 50.16 for precision,

recall and F-score in Fig 10 but the displayed Binary score for SCRQE in

Fig 11 is 47.73. What does it denote?

The description of the Inter-Annotator Agreement (IAA) analysis is greatly

extended in the current revision, however, it also showed a major

discrepancy in the way how this measure was used. The main reason for

measuring IAA lies in offering an estimate of how "difficult" the annotated

(sub)tasks are for human annotators as an approximate comparison to the

system performance. IAA must thus be measured on independently annotated

datasets without any kind of post-annotation iterative processing,

discussions and refinements of the measured data. Here, the authors have

used IAA for a different purpose, i.e. for reducing the disagreement in the

final dataset. What is also still not clear from the text is whether each

of the three annotators has processed all 1275 dataset questions.

In lines 1324-1336, the authors explain imprecise usage of the intersection

operation applied to non-sets as a "metaphorical interpretation of

intersection". However, equations are to provide exactness to textual

descriptions, not metaphors. A correct way is to adjust the equations to be

mathematically sound. It may be seen that the flawed equations propagate

from the original paper (Liu et al, 2021), which is, however, not cited in

this section. A fixed interpretation of these equations may be found in

another follow-up paper

Q. Xu et al., "GCN-based End-to-End Model for Comparative Opinion

Quintuple Extraction," 2023 International Joint Conference on Neural

Networks (IJCNN), Gold Coast, Australia, 2023, pp. 1-6, doi:

10.1109/IJCNN54540.2023.10191436.

where the intersection is applied to the quintuples, not to their elements.

The main difference is that "g_i indicates the i−th gold answer quintuple,

p_i denotes the i−th predicted quintuple result" (Xu et al, 2023, correct)

versus "p_i and g_i represent the i-th element in the predicted and gold

quintuple comparative relations" (both Liu et al, 2021 and the current

article, incorrect).

In lines 107-109 and repeated further, the Fleiss' Kappa is expressed as

percentages, which does not correspond to the definition of this measure

which reaches values in the interval from -1 to 1.

Table 17 compares the element extraction task results for three variants of

the implementation with bold numbers for the MTL+adapter model even in

cases where there are better results for the other variants. Why?

Errors in text:

line 225, "RoBERTa, which is finely tuned" -> "... is fine-tuned"

line 743: ", where:" but nothing follows

line 845, wrong section number 3.2.2.1. after 3.3.2

lines 1081-1084, the same statement twice:

- "categories ... either leave the direction of preference or the roles

of the entities ambiguous or undefined."

- "In these categories, one or more of these elements—either the

direction of preference or the roles of the entities—are typically

left ambiguous or undefined."

Reviewer #2: Concerns are as follows:

1. The methodological level of the narrative in the abstract be shortened to focus more on the problem and innovation.

2. Innovations are too broad, and metrics and data sets alone are insufficient as a fundamental basis for innovation.

3. SCQRD is a redundancy about the methodology, and irrelevant content needs to be removed, e.g., datasets, traditional methods, performance metrics.

4. There is a need for an overall block diagram of the SCQRD rather than a textual representation of the components.

5. There is a need for updating references.

7. PLOS authors have the option to publish the peer review history of their article (what does this mean? ). If published, this will include your full peer review and any attached files.

**Do you want your identity to be public for this peer review?** For information about this choice, including consent withdrawal, please see our Privacy Policy .

Reviewer #1: No

Reviewer #2: No

---

## [Author Response · Author response to Decision Letter 2]

4 Nov 2024

View Letter

Date: Jun 11 2024 09:30PM

To: "Afsaneh Fatemi" a_fatemi@eng.ui.ac.ir

From: "PLOS ONE" plosone@plos.org

Subject: PLOS ONE Decision: Revision required [PONE-D-23-39359R1]

PONE-D-23-39359R1

SCRQE: Subjective Comparative Relation Quintuple Extraction from Questions in Product Domain

PLOS ONE

Dear Dr. Fatemi,

Thank you for submitting your manuscript to PLOS ONE. After careful consideration, we feel that it has merit but does not fully meet PLOS ONE’s publication criteria as it currently stands. Therefore, we invite you to submit a revised version of the manuscript that addresses the points raised during the review process.

We look forward to receiving your revised manuscript.

Kind regards,

Leona Cilar Budler

Academic Editor

PLOS ONE

Response to Editor's Invitation for Manuscript Revision

Dear Dr. Leona Cilar Budler,

We sincerely appreciate the thoughtful evaluation of our manuscript and the opportunity to revise and resubmit it for consideration in PLOS ONE. The constructive comments and suggestions from the review process are invaluable, and we are confident that addressing them will significantly enhance the clarity and impact of our research.

After carefully reviewing the feedback, we have made substantial revisions to our manuscript, focusing particularly on improving methodological clarity, providing stronger context and justification for our work, and incorporating more relevant comparisons with recent advancements in the field, as advised.

In line with your submission guidelines, we are prepared to submit the revised manuscript by the provided deadline, along with the following documents:

Rebuttal Letter: A detailed response addressing each point raised by the academic editor and reviewers, which will be uploaded as a separate file labeled 'Response to Reviewers.'

Marked-Up Manuscript: A version of the manuscript with all changes clearly tracked for ease of review, to be submitted as 'Revised Manuscript with Track Changes.'

Unmarked Manuscript: A clean version of the revised manuscript without tracked changes, labeled 'Manuscript.'

Additionally, we are pleased to inform you that our datasets and code are available on GitHub at: https://github.com/mahsamb/SCRQD. We believe this transparency will contribute to the reproducibility of our research.

We will also revisit our financial disclosure statement and update it, if necessary, in the cover letter.

Regarding the recommendation to deposit our laboratory protocols on protocols.io, we recognize the importance of promoting reproducibility and transparency in research. We plan to deposit our protocols accordingly and contribute to these best practices in scientific research.

We are excited to move forward with the revision process and are grateful for the guidance and support provided by PLOS ONE. Should you require any additional information or clarification, please do not hesitate to contact us.

Warm regards,

Dr. Afsaneh Fatemi

Additional Editor Comments:

There are some suggestions and comments from reviewers to improve your paper. Please take them into account to improve your paper.

Response to Editor's Additional Comments:

Dear Editor,

Thank you for your additional comments regarding the reviewer feedback. We appreciate the suggestions provided by the reviewers, and we will thoroughly address each point in our revised manuscript. We are committed to ensuring that all the recommendations are taken into account to improve the quality and clarity of our paper.

We will provide a detailed response to each of the reviewers' comments in the rebuttal letter, alongside the revised manuscript, as per the submission guidelines.

Once again, thank you for your guidance and for the opportunity to enhance our manuscript.

Warm regards,

Dr. Afsaneh Fatemi

Reviewers' comments:

Reviewer's Responses to Questions

Opening Response to the Reviewers:

Dear Reviewers,

Thank you for your insightful comments and constructive critique. We sincerely appreciate your thorough review and the opportunity to refine the sections you highlighted for improvement. Our commitment lies in enhancing the quality and transparency of our manuscript, and we have made revisions accordingly based on your valuable suggestions.

Additionally, we are pleased to inform you that our datasets and code are available on GitHub at: https://github.com/mahsamb/SCRQD. We believe this transparency will contribute to the reproducibility of our research.

Warm regards,

Dr. Afsaneh Fatemi

Comments to the Author

1. If the authors have adequately addressed your comments raised in a previous round of review and you feel that this manuscript is now acceptable for publication, you may indicate that here to bypass the “Comments to the Author” section, enter your conflict of interest statement in the “Confidential to Editor” section, and submit your "Accept" recommendation.

Reviewer #1: (No Response)

Reviewer #2: All comments have been addressed

2. Is the manuscript technically sound, and do the data support the conclusions?

Reviewer #1: Partly

Reviewer #2: Yes

3. Has the statistical analysis been performed appropriately and rigorously?

Reviewer #1: N/A

Reviewer #2: N/A

4. Have the authors made all data underlying the findings in their manuscript fully available?

Reviewer #1: Yes

Reviewer #2: Yes

5. Is the manuscript presented in an intelligible fashion and written in standard English?

Reviewer #1: Yes

Reviewer #2: Yes

6. Review Comments to the Author

Reviewer #1: The authors have provided a substantial revision of the original text, adding

a number of previously missing details of the work. While the new text has

significantly improved the critical points of the authors' work, several

new questions arise accompanying the added details.

Addressing Reviewer Feedback: Clarifications and Revisions

Dear Reviewer,

Thank you for your thorough review and valuable feedback. We sincerely appreciate the time and effort you have invested in evaluating our work. Your insights have provided us with an invaluable opportunity to refine our manuscript and clarify key aspects of our research. We are grateful for the chance to address your concerns and to offer further clarification on our motivations, methodologies, and the impact of our proposed changes. Below are our detailed responses to each of your comments, along with explanations of the revisions we have made to enhance the clarity, accuracy, and overall quality of our work.

One of the crucial points lies in the subject difference to most existing

datasets, i.e. to working with comparative questions instead of comparative

sentences. The authors point at the differences (and complications) between

these two data approaches, but a motivation for processing questions is

missing. A possible reason could be connected with "answers", that are

natural companions of questions, but these are not touched at all in the

text.

Dear Reviewer,

Thank you for your insightful comments. We appreciate you highlighting the need for clearer motivation regarding our focus on comparative questions and the connection to answer generation. We have revised the manuscript to address these points.

Motivation for Processing Questions Instead of Sentences

The motivation for focusing on subjective comparative questions rather than sentences is rooted in the way users naturally interact with systems, particularly in e-commerce platforms and question-answering applications. In these contexts, users often pose comparative questions when they are in the process of making decisions, such as choosing between products or services. These questions typically seek to draw direct comparisons between two or more entities, reflecting the user's intent to evaluate specific features or aspects that are important to their decision-making process.

Comparative questions are more than just statements; they capture the active search for answers that keeps users engaged. For instance, when a user asks, "Is Product A better than Product B in terms of quality?" they are not merely stating a fact but are actively seeking information that will influence their purchase decision. This interaction is inherently dynamic and driven by specific goals, making the analysis of questions more pertinent to real-world applications compared to analyzing standalone sentences.

Our model, by concentrating on the analysis of questions, better captures user intent—preferences, priorities, and context—leading to more accurate and tailored responses. This approach not only improves user satisfaction but also serves as an essential initial step in the automatic question answering (AQA) process. Additionally, subjective comparative questions feature distinct comparative preference classifications (CPC) compared to sentences, further justifying the need for this focused, question-driven analysis.

In summary, focusing on subjective comparative questions aligns our model with real-world user interactions and aids in the development of more refined question-answering systems.

The Role of Question Analysis in Answering Subjective Comparative Questions

You are correct that in our initial manuscript, we did not fully explain the role of question analysis in answering subjective comparative questions. However, we have now addressed this in our revised manuscript by including a detailed discussion in the Introduction section. In our revised manuscript, we provide a comprehensive analysis of the role of question analysis in answering subjective comparative questions. Specifically, we focus on the detailed analysis of comparative questions as a crucial step in the automatic question answering (AQA) process, which involves identifying and classifying key elements such as subject and object entities, compared aspects, comparative preferences, and constraints. By accurately determining these elements, our model ensures that answers are relevant, contextually precise, and aligned with the user's intent, ultimately assisting users in making informed decisions. This foundational analysis is essential for generating meaningful and actionable answers in subsequent stages of AQA, and provides valuable insights that can be leveraged by other research efforts aimed at answer generation. By incorporating these elements into our analysis, we ensure that the insights generated can be effectively used by other systems to produce meaningful, actionable answers that align with user preferences and improve decision-making.

To honor the reviewer's input and ensure the thoroughness of our response, we have expanded the Introduction in our revised manuscript (lines 71-128) to provide a detailed discussion on the motivation for focusing on comparative questions rather than sentences:

Our model's outputs, particularly the determination of subject and object entities, the identification of compared aspects, the classification of comparative preferences (CPC), and the extraction of constraints, play a pivotal role in shaping the content and form of the answers:

Determining Subject/Object Entities: By accurately identifying the entities being compared, our model ensures that the answer is directly relevant to the user's query, focusing on the correct entities without ambiguity.

Compared Aspects: The identification of specific aspects under comparison allows the generated answer to be precise and contextually relevant, addressing exactly what the user is interested in, such as performance, design, or price.

Comparative Preference Classification (CPC) of Questions: The CPC process categorizes the nature of the comparison within the question, such as whether one entity is "better," "worse," or "equal" to another, or more nuanced preferences like "significantly better" or "marginally worse." This classification profoundly influences the answer generated in response to a subjective comparative question, as it directly shapes the direction, tone, and content of the answer.

Alignment of Answer Tone with Question’s Intent: By identifying the specific comparative preference expressed in the question, the CPC helps ensure that the answer's tone aligns with the user’s intent. For example, if a question asks whether a product is "significantly better" than another, the answer can affirm this with strong, decisive language if the comparison holds true, or gently refute it with explanations if it does not. This alignment is critical in providing an answer that resonates with the user’s expectations

---

## [Decision Letter · Decision Letter 2]

10 Jan 2025

PONE-D-23-39359R2SCRQE: Subjective Comparative Relation Quintuple Extraction from Questions in Product DomainPLOS ONE

Dear Dr. Fatemi,

Thank you for submitting your manuscript to PLOS ONE. After careful consideration, we feel that it has merit but does not fully meet PLOS ONE’s publication criteria as it currently stands. Therefore, we invite you to submit a revised version of the manuscript that addresses the points raised during the review process.

We look forward to receiving your revised manuscript.

Kind regards,

Leona Cilar Budler

Academic Editor

PLOS ONE

**Journal Requirements:**

**Additional Editor Comments:**

There are some minor issues listed that authors need to resolve to consider this paper for publication.

Reviewers' comments:

Reviewer's Responses to Questions

**Comments to the Author**

1. If the authors have adequately addressed your comments raised in a previous round of review and you feel that this manuscript is now acceptable for publication, you may indicate that here to bypass the “Comments to the Author” section, enter your conflict of interest statement in the “Confidential to Editor” section, and submit your "Accept" recommendation.

Reviewer #1: All comments have been addressed

Reviewer #2: All comments have been addressed

2. Is the manuscript technically sound, and do the data support the conclusions?

Reviewer #1: Yes

Reviewer #2: Yes

3. Has the statistical analysis been performed appropriately and rigorously? 

Reviewer #1: Yes

Reviewer #2: Yes

4. Have the authors made all data underlying the findings in their manuscript fully available?

Reviewer #1: Yes

Reviewer #2: Yes

5. Is the manuscript presented in an intelligible fashion and written in standard English?

Reviewer #1: Yes

Reviewer #2: Yes

6. Review Comments to the Author

**Reviewer #1: ** The authors have again substantially extended the article text (56 pages vs

32 pages in version 1) with detailed reactions to reviewers comments and

further work of the authors. The presented SCQRD dataset has been expanded

almost three times offering its enlarged version denoted as

Smartphone-SCQRE and a version adapted from the public CompSent-19 dataset

named here as the Brands-CompSent-19-SCQRE dataset. This has led to

improved coverage of previously underrepresented question types. Both

datasets have undergone a newly adapted annotation process with more

annotators and improved Inter-Annotator Agreement evaluation.

In the previous round, the authors have based the model architecture on the

RoBERTa base model. In the current text, the proposed system exploits

a specific model of RoBERTa_base_go_emotions, which is mentioned more than

40 times in the text (even used as one of the article keywords) but not

properly introduced until Section 3.3.1 at page 22. Moreover, the citation

with the model is not a correct one, as it refers to the original RoBERTa

model (without fine-tuning for emotions). If the model choice is so

crucial, for the task, it should be briefly introduced near its first

mention.

The RoBERTa_base_go_emotions model is evaluated together with newer

generative models (GPT-3.5-turbo-0613, Llama 2 70B Chat, and Qwen 1.5 7B

Chat) that were employed using their native prompt-based tasks. However,

for the purpose of Text Classification (like RoBERTa_base_go_emotions), the

generative models can be standardly fine-tuned with extra classification

layer(s). This usually offers better results than just prompting. But even

evaluating the prompt-based approaches (with the level of details as

offered in the article) is a valuable experiment.

The presented SCRQE model has been newly trained with the extended datasets

and with augmented reversed comparative statements which improved the

generalization capabilities and the resulting scores of the proposed model

so that it now outperforms a previously better model.

Overall, I believe the article is acceptable, with the correction of the

late introduction of the RoBERTa_base_go_emotions model.

**Reviewer #2:**  (No Response)

7. PLOS authors have the option to publish the peer review history of their article (what does this mean? ). If published, this will include your full peer review and any attached files.

**Do you want your identity to be public for this peer review?** For information about this choice, including consent withdrawal, please see our Privacy Policy .

Reviewer #1: No

Reviewer #2: No

---

## [Author Response · Author response to Decision Letter 3]

17 Jan 2025

View Letter

Date: Jan 10 2025 06:13AM

To: "Afsaneh Fatemi" a_fatemi@eng.ui.ac.ir

From: "PLOS ONE" plosone@plos.org

Subject: PLOS ONE Decision: Revision required [PONE-D-23-39359R2]

PONE-D-23-39359R2

SCRQE: Subjective Comparative Relation Quintuple Extraction from Questions in Product Domain

PLOS ONE

Dear Dr. Fatemi,

Thank you for submitting your manuscript to PLOS ONE. After careful consideration, we feel that it has merit but does not fully meet PLOS ONE’s publication criteria as it currently stands. Therefore, we invite you to submit a revised version of the manuscript that addresses the points raised during the review process.

We look forward to receiving your revised manuscript.

Kind regards,

Leona Cilar Budler

Academic Editor

PLOS ONE

Response to Revision Request for Manuscript ID [PONE-D-23-39359R2]

Dear Dr. Leona Cilar Budler,

Thank you for your detailed feedback and the opportunity to revise our manuscript titled "SCRQE: Subjective Comparative Relation Quintuple Extraction from Questions in Product Domain," with ID [PONE-D-23-39359R2]. We greatly appreciate the constructive comments from the reviewers and recognize the significant value they add to our study.

We are diligently addressing the points raised during the review process and are confident that our revisions will meet PLOS ONE’s publication criteria. We aim to submit the revised manuscript by the set deadline of February 24, 2025. Should we require additional time to complete our revisions, we will notify you promptly.

As requested, we will include the following items in our submission:

1. Rebuttal Letter: A document labeled 'Response to Reviewers', addressing each point raised by the academic editor and reviewers.

2. Marked-Up Copy of the Manuscript: This document will highlight changes made to the original version and will be labeled 'Revised Manuscript with Track Changes'.

3. Unmarked Version of the Revised Paper: This document will be uploaded as 'Manuscript' and will not contain tracked changes.

Additionally, we have included a comprehensive title page within the main manuscript document, listing all authors, their affiliations, and clearly indicating the corresponding author, as per PLOS ONE’s author instructions. All references have been meticulously converted to Vancouver style and checked for accuracy, resulting in the addition of References [4] and [6], the replacement of the former Reference [22] with Reference [25], and the renumbering of existing references to maintain consistency and adherence to the journal’s formatting guidelines. In response to Reviewer #1, we have introduced the RoBERTa_base_go_emotions model earlier in the manuscript, corrected its citation to include the fine-tuned GoEmotions resource, and clarified our focus on prompt-based evaluations over fine-tuning generative models. Specifically, while these generative models can indeed be fine-tuned with dedicated classification layers, we chose to focus on prompt-based evaluations to provide a straightforward zero-/few-shot comparison across different model families, with fine-tuning being a promising direction for future work. These changes enhance the clarity and quality of our manuscript, aligning it with the journal’s requirements.

We will also review our financial disclosure statement and update it if necessary. Although our study does not involve laboratory protocols, we have ensured that our data preparation, model implementation, and experimental setups are described in sufficient detail to ensure reproducibility. Additionally, all datasets and relevant code have been made publicly available as stated in our Data Availability section.

Thank you once again for considering our work. We look forward to the opportunity to enhance our manuscript and contribute to PLOS ONE.

Sincerely,

Afsaneh Fatemi

Corresponding Author on behalf of all co-authors

Journal Requirements:

Response to the Editor

Dear Editor,

We appreciate your careful review of our manuscript and the opportunity to address the journal’s requirements. Below, please find our point-by-point response regarding our updated reference list:

1. Review and Update of Reference List

o We have thoroughly examined our entire reference list to ensure it is complete, accurate, and current.

o No Retracted References: We confirm that none of the sources cited in our revised manuscript are retracted. However, we replaced the former Reference 22 with a new Reference 25 to include a more current and relevant source.

o Conversion to Vancouver Style: All references have been converted to Vancouver style and have been checked for accuracy to ensure they adhere to the journal’s formatting requirements.

o Newly Added References: We have introduced three new references in this revision:

1. References 4 and 6: These pertain to the roberta-base_go_emotions model and the GoEmotions dataset, respectively. Initially, we only included a Hugging Face footnote for SamLowe/roberta-base-go_emotions, but we now explicitly cite both the specialized model and GoEmotions to clarify the lineage from RoBERTa to the fine-tuned variant.

2. Reference 25: This replaces the previous Reference 22 in the earlier version of our manuscript. We opted for a more recent source that provides updated findings relevant to our research topic.

o As a result of adding these references, the numbering of subsequent items in the list has shifted accordingly. Additionally, Reference 30 in the previous manuscript has been moved forward to Reference 5 as it is now mentioned earlier in the revised manuscript.

2. Rationale for Retractions or Replacements

o As noted above, no citations in our manuscript are retracted; thus, we did not need to provide any retraction notices.

o The only replacement involves the introduction of Reference 25 in place of the former Reference 22, due to its more comprehensive and up-to-date information.

3. Mention of Reference List Changes in Rebuttal Letter

o In accordance with the journal’s guidelines, we confirm here that our only changes involve adding these three new references (4, 6, and 25), thereby causing a renumbering of some existing entries.

o References 4 & 6: In Section 3.3.1, we clarify the fine-tuning lineage by citing the original RoBERTa paper [Liu et al., 2019], the SamLowe/roberta-base-go_emotions model (including its Hugging Face link), and the GoEmotions dataset [Demszky et al., 2020]. We specifically highlight how the roberta-base_go_emotions variant differs from base RoBERTa in its multi-label emotion classification capabilities.

o Reference 25: This reference replaces the old 22 in our earlier manuscript and reflects a more updated source with findings that are directly applicable to our study.

o Additional Renumbering: Due to the earlier mention in the revised text, the previous Reference 30 now appears as Reference 5 in the new manuscript.

o Conversion to Vancouver Style: We have converted all references to Vancouver style, ensuring consistency and adherence to the journal’s formatting guidelines. Each reference has been thoroughly checked for correctness.

Additional Notes for Clarity

• Reference Additions and Replacements:

o References 4 and 6 were added to provide direct citations for the roberta-base_go_emotions model and the GoEmotions dataset, respectively. These additions clarify the model’s lineage and its relevance to our study.

o Reference 25 was introduced to replace Reference 22 with a more recent and relevant source that enhances the credibility and up-to-dateness of our research.

• Conversion to Vancouver Style:

o All references have been meticulously converted to Vancouver style, adhering to the journal’s specific formatting guidelines. This includes proper author formatting, title capitalization, journal abbreviations, and accurate page numbering.

• Renumbering Impact:

o The addition of new references 4 and 6, along with the replacement of 22 with 25, has resulted in a systematic renumbering of subsequent references. For example, the previous Reference 30 is now listed as Reference 5 due to its earlier mention in the revised manuscript.

We have ensured that these changes enhance the clarity and accuracy of our manuscript, aligning it with the journal’s standards and requirements. We trust that these revisions fulfill the journal’s requirements, and we remain available to provide any additional clarifications or materials as necessary. Once again, we appreciate your guidance and consideration.

To honor the reviewer's input and ensure the thoroughness of our response, we have included references to both the previous version of the manuscript and the current revision in our response letter:

Previous version of the manuscript:

6. References

1. Moghaddam, S. and M. Ester. AQA: aspect-based opinion question answering. in 2011 IEEE 11th International Conference on Data Mining Workshops. 2011. IEEE.

2. Yu, J., Z.-J. Zha, and T.-S. Chua. Answering opinion questions on products by exploiting hierarchical organization of consumer reviews. in Proceedings of the 2012 joint conference on empirical methods in natural language processing and computational natural language learning. 2012.

3. Bayoudhi, A., H. Ghorbel, and L. Hadrich Belguith. Question answering system for dialogues: A new taxonomy of opinion questions. in Flexible Query Answering Systems: 10th International Conference, FQAS 2013, Granada, Spain, September 18-20, 2013. Proceedings 10. 2013. Springer.

4. Pavlopoulos, I., Aspect based sentiment analysis. Athens University of Economics and Business, 2014.

5. Liu, Z., R. Xia, and J. Yu. Comparative opinion quintuple extraction from product reviews. in Proceedings of the 2021 conference on empirical methods in natural language processing. 2021.

6. Panchenko, A., et al., Categorizing comparative sentences. arXiv preprint arXiv:1809.06152, 2018.

7. Li, S., et al., Comparable entity mining from comparative questions. IEEE transactions on knowledge and data engineering, 2011. 25(7): p. 1498-1509.

8. REDDY, V. and B. KMahesh, A Method for Comparable Mining from Comparative Questions. 2014.

9. Saelan, A., A. Purwarianti, and D. Widyantoro. Question analysis for Indonesian comparative question. in Journal of Physics: Conference Series. 2017. IOP Publishing.

10. Saelan, A., A. Purwarianti, and D.H. Widyantoro, Answering Comparison in Indonesian Question Answering System with Database. International Journal on Electrical Engineering & Informatics, 2018. 10(4).

11. Liu, Z., C.-X. Qin, and Y.-J. Zhang, Mining product competitiveness by fusing multisource online information. Decision Support Systems, 2021. 143: p. 113477.

12. Jindal, N. and B. Liu. Mining comparative sentences and relations. in Aaai. 2006.

13. Xu, K., et al., Mining comparative opinions from customer reviews for competitive intelligence. Decision support systems, 2011. 50(4): p. 743-754.

14. Arora, J., et al. Extracting entities of interest from comparative product reviews. in Proceedings of the 2017 ACM on Conference on Information and Knowledge Management. 2017.

15. Yang Z, Xu F, Yu J, Xia R, editors. UniCOQE: Unified comparative opinion quintuple extraction as a set. Findings of the Association for Computational Linguistics: ACL 2023; 2023.

16. Gao, F., et al., End-to-end comparative opinion quintuple extraction as bipartite set prediction with dynamic structure pruning. Expert Systems with Applications, 2024. 245: p. 123058.

17. Wang, H., et al., Competitiveness analysis through comparative relation mining: evidence from restaurants’ online reviews. Industrial Management & Data Systems, 2017. 117(4): p. 672-687.

18. Liu, D., L. Wang, and Y. Shao. Multi-task learning neural networks for comparative elements extraction. in Chinese Lexical Semantics: 21st Workshop, CLSW 2020, Hong Kong, China, May 28–30, 2020, Revised Selected Papers 21. 2021. Springer.

19. Younis, U., et al., Applying machine learning techniques for performing comparative opinion mining. Open Computer Science, 2020. 10(1): p. 461-477.

20. Kang I, Ruan S, Ho T, Lin J-C, Mohsin F, Seneviratne O, et al. LLM-augmented Preference Learning from Natural Language. arXiv preprint arXiv:231008523. 2023.

21. Ma, N., et al. Entity-aware dependency-based deep graph attention network for comparative preference classification. in Proceedings of Annual Meeting of the Association for Computational Linguistics (ACL-2020). 2020.

22. Kang, Y., A Hierarchical Framework for Online Product Review Helpfulness Assessment. 2017.

23. Cruz, I., A.F. Gelbukh, and G. Sidorov, Implicit Aspect Indicator Extraction for Aspect based Opinion Mining. Int. J. Comput. Linguistics Appl., 2014. 5(2): p. 135-152.

24. Babaali M, Fatemi A, Nematbakhsh MA. Creating and validating the Fine-Grained Question Subjectivity Dataset (FQSD): A new benchmark for enhanced automatic subjective question answering systems. Plos one. 2024;19(5):e0301696.

25. Augustyniak, Ł., T. Kajdanowicz, and P. Kazienko. Aspect detection using word and char embeddings with (Bi) LSTM and CRF. in 2019 IEEE second international conference on artificial intelligence and knowledge engineering (AIKE). 2019. IEEE.

26. Fleiss, J.L., Measuring nominal scale agreement among many raters. Psychological bulletin, 1971. 76(5): p. 378.

27. Cohen, J., Weighted kappa: Nominal scale agreement provision for scaled disagreement or partial credit. Psychological bulletin, 1968. 70(4): p. 213.

28. Krippendorff, K., Reliability in content analysis: Some common misconceptions and recommendations. Human communication research, 2004. 30(3): p. 411-433.

29. Krippendorff, K., Validity in content analysis, in Computerstrategien für die Kommunikationsanalyse. 1980.

30. Liu, Y., et al., Roberta: A robustly optimized bert pretraining approach. arXiv preprint arXiv:1907.11692, 2019.

---

## [Decision Letter · Decision Letter 3]

9 Feb 2025

SCRQE: Subjective Comparative Relation Quintuple Extraction from Questions in Product Domain

PONE-D-23-39359R3

Dear Dr. Fatemi,

We’re pleased to inform you that your manuscript has been judged scientifically suitable for publication and will be formally accepted for publication once it meets all outstanding technical requirements.

Kind regards,

Leona Cilar Budler

Academic Editor

PLOS ONE

Additional Editor Comments (optional):

No further comments

Reviewers' comments:

Reviewer's Responses to Questions

**Comments to the Author**

1. If the authors have adequately addressed your comments raised in a previous round of review and you feel that this manuscript is now acceptable for publication, you may indicate that here to bypass the “Comments to the Author” section, enter your conflict of interest statement in the “Confidential to Editor” section, and submit your "Accept" recommendation.

Reviewer #1: All comments have been addressed

Reviewer #2: All comments have been addressed

2. Is the manuscript technically sound, and do the data support the conclusions?

Reviewer #1: Yes

Reviewer #2: Yes

3. Has the statistical analysis been performed appropriately and rigorously? 

Reviewer #1: Yes

Reviewer #2: Yes

4. Have the authors made all data underlying the findings in their manuscript fully available?

Reviewer #1: Yes

Reviewer #2: Yes

5. Is the manuscript presented in an intelligible fashion and written in standard English?

Reviewer #1: Yes

Reviewer #2: Yes

6. Review Comments to the Author

Reviewer #1: In the preceding round, the only suggestions proposed were the introduction

of the RoBERTa_base_go_emotions model and a mention of the fine-tuning

approach for generative models as future work. Both of these suggestions

are addressed in the text, therefore, the article is considered acceptable.

Reviewer #2: (No Response)

7. PLOS authors have the option to publish the peer review history of their article (what does this mean? ). If published, this will include your full peer review and any attached files.

**Do you want your identity to be public for this peer review?** For information about this choice, including consent withdrawal, please see our Privacy Policy .

Reviewer #1: No

Reviewer #2: **Yes: ** weibing wan

---

## [Editor Report · Acceptance letter]

PONE-D-23-39359R3

PLOS ONE

Dear Dr. Fatemi,

I'm pleased to inform you that your manuscript has been deemed suitable for publication in PLOS ONE. Congratulations! Your manuscript is now being handed over to our production team.

Kind regards,

on behalf of

Dr. Leona Cilar Budler

Academic Editor

PLOS ONE